# Multiplexed screening reveals how cancer-specific alternative polyadenylation shapes tumor growth in vivo

Austin M. Gabel [1,2,3,4], Andrea E. Belleville [1,2,4,5], James D. Thomas [1,2], Siegen A. McKellar [1,2,4,5], Taylor R. Nicholas[1,2], Toshihiro Banjo [1,2], Edie I. Crosse [1,2] & Robert K. Bradley [1,2,3] ✉

Alternative polyadenylation (APA) is strikingly dysregulated in many cancers. Although global APA dysregulation is frequently associated with poor prognosis, the importance of most individual APA events is controversial simply because few have been functionally studied. Here, we address this gap by developing a CRISPR-Cas9-based screen to manipulate endogenous polyadenylation and systematically quantify how APA events contribute to tumor growth in vivo. Our screen reveals individual APA events that control mouse melanoma growth in an immunocompetent host, with concordant associations in clinical human cancer. For example, forced *Atg7* 3′ UTR lengthening in mouse melanoma suppresses ATG7 protein levels, slows tumor growth, and improves host survival; similarly, in clinical human melanoma, a long *ATG7* 3′ UTR is associated with significantly prolonged patient survival. Overall, our study provides an easily adaptable means to functionally dissect APA in physiological systems and directly quantifies the contributions of recurrent APA events to tumorigenic phenotypes.

Alternative cleavage and polyadenylation (APA)—the process whereby a pre-mRNA can be cleaved and have a poly(A) tail added at multiple distinct locations, leading to expression of mRNAs with different 3′ untranslated regions (UTRs)—affects most human genes[1]. Although the biological roles of most APA events are unknown, detailed studies have revealed that APA can affect mRNA levels, localization, and translation, among other molecular phenotypes[2–4]. Poly(A) site selection is coordinated by differential interactions between multiple factors, including several RNA-binding proteins and the U1 small nuclear RNA (snRNA). RNA-binding proteins and the U1 snRNA bind RNAs in a sequence-specific manner near the proximal poly(A) site and can either promote or inhibit assembly of the 3′ end processing complex, although this process is still not entirely understood[5–8]. Differential polyadenylation site usage is commonly observed in many biological processes, frequently in a biased manner indicative of coordinated regulatory changes[9–11]. For example, rapidly dividing cells frequently utilize gene-proximal over gene-distal poly(A) sites, and thus express mRNAs with correspondingly shorter 3′ UTRs, compared to non-dividing, terminally differentiated cells. mRNAs are differentially polyadenylated throughout development, where terminally differentiated cells tend to utilize more distal poly(A) sites[3]; immune cell subsets including monocytes, T cells, and B cells undergo global 3′ UTR shortening when stimulated by their respective chemokines to begin dividing[12]; many cancers express mRNAs with markedly shorter 3′ UTRs than do peritumoral, healthy tissues[13–15].

It is well-established that APA is recurrently dysregulated in virtually all cancer types, yet it remains unknown whether most APA

[1]Computational Biology Program, Public Health Sciences Division, Fred Hutchinson Cancer Center, Seattle, WA, USA. [2]Basic Sciences Division, Fred Hutchinson Cancer Center, Seattle, WA, USA. [3]Department of Genome Sciences, University of Washington, Seattle, WA, USA. [4]Medical Scientist Training Program, University of Washington, Seattle, WA, USA. [5]Molecular and Cellular Biology Program, University of Washington, Seattle, WA, USA. ✉e-mail: rbradley@fredhutch.org

dysregulation directly promotes cancer progression or instead simply a downstream consequence of rapid cell division[16–20]. Consistent with a functional role for APA dysregulation in cancers, several studies have reported that transcriptome-wide APA dysregulation is associated with patient prognosis in a subset of tumor types and that overexpression of a specific mRNA with a short 3′ UTR, but not a long 3′ UTR, can promote enhanced cancer cell growth[21,22]. Nonetheless, the potential functional relevance and roles of cancer-associated APA remains unclear. The vast majority of cancer-associated APA events have never been functionally studied. Most functional studies of APA have relied on transgenic expression, rather than manipulation of APA in an endogenous context, and no study has yet manipulated endogenous, cancer-associated APA events in the physiological context of tumorigenesis in vivo.

Here, we systematically identify cancer-associated APA dysregulation that plays a causative, rather than simply correlative, role in tumorigenesis by functionally studying endogenous APA site selection in the physiologically relevant setting of tumorigenesis in vivo. To do so, we developed a multiplexed, CRISPR-Cas9-based screen that permitted us to perform parallelized interrogation of individual APA events during tumorigenesis in vivo. This high-throughput functional screen revealed APA events that control key tumor phenotypes in an immunocompetent mouse host and have concordant associations in clinical human cancer.

## Results

### Global 3′ UTR lengthening predicts poor patient prognosis in human melanoma

To assess whether cancer-associated changes in APA are relevant to clinical phenotypes, we tested whether global APA dysregulation was significantly associated with patient outcomes. Several prior studies have analyzed RNA sequencing data from The Cancer Genome Atlas (TCGA) and observed significant associations between global APA dysregulation and patient survival; however, those studies limited analyses to the 17 tumor types for which there were sufficient patient-matched, peritumoral normal tissue samples available[13,15]. In order to extend such analyses to all tumor types, including those for which peritumoral samples are not available, we instead took a stratification-based approach that relied on data from tumor samples alone. For each cancer subtype, we stratified patients into terciles representing whether their tumor transcriptomes preferentially expressed short, medium, or long 3′ UTRs by computing a median 3′ UTR length for each tumor across 7513 genes that are subject to alternative polyadenylation (Fig. 1A–H; Supplementary Fig. 1A–C). This measure of global 3′ UTR length was significantly positively correlated with *PABPN1* expression in 28 of 30 cancer subtypes and significantly negatively correlated with *CSTF2* expression in 21 of 30 cancer subtypes, consistent with *PABPN1* and *CSTF2*'s known roles as a repressor or activator of proximal poly(A) site usage, respectively (Supplementary Fig. 1D–I)[5–7].

We then performed Kaplan–Meier survival analysis by comparing patients with short or long median 3′ UTRs and found significant survival differences that varied substantially by cancer type (Fig. 1I; Supplementary Fig. 2A–D; Supplementary Data File 1). In ovarian carcinoma, renal cell carcinoma, breast carcinoma, colon adenocarcinoma, and lung adenocarcinoma, patients whose cancers expressed shorter 3′ UTRs exhibited significantly worse overall survival (Fig. 1J; Supplementary Fig. 2D). Several cancer subtypes displayed the opposite trend, where patients with globally lengthened 3′ UTRs exhibited significantly worse overall survival, including head and neck squamous cell carcinoma, low grade glioma, and cutaneous melanoma (Fig. 1I, J; Supplementary Fig. 2D). These variable associations illustrate the potentially clinically relevant roles of APA and highlight the likely complexity of the relationship between 3′ UTR length trends and prognosis.

These results are necessarily based upon computational inference of 3′ UTR length using poly(A)-selected RNA-seq data, rather than assays like 3′-seq that provide nucleotide-level resolution of 3′ UTR length, due to the nature of most patient transcriptomic data. We therefore confirmed that our results were agnostic to the specific computational algorithm used to infer 3′ UTR usage by measuring 3′ UTR length using a distinct method, APAlyzer[23]. We repeated our analyses for the cutaneous melanoma cohort, which demonstrated the most significant TCGA associations between 3′ UTR length and patient survival among TCGA cohorts, and found concordant and similarly strong associations between 3′ UTR length and survival (Supplementary Fig. 2E, F; Supplementary Data File 2). Together, these analyses highlight a subset of cancer types where global APA is strongly correlated with patient survival, suggesting possible functional relationships between APA and tumor growth.

### A mouse model of melanoma exhibits cancer-associated aberrant alternative polyadenylation

Motivated by these strong associations between APA and patient prognosis, we sought to experimentally test the hypothesis that APA dysregulation affecting specific genes functionally contributes to tumorigenesis. As clinical melanoma exhibited the strongest correlations between 3′ UTR length and patient survival (Fig. 1I, J), we turned to syngeneic mouse models of untransformed melanocytes and melanoma to functionally study APA in biologically relevant and experimentally tractable systems. B16-F10 cells are mouse melanoma cells that readily engraft and form aggressive tumors in immunocompetent C57BL/6 mice, while Melan-A cells are immortalized but non-tumorigenic melanocytes derived from the same C57BL/6 background (Fig. 2A)[24,25]. These two models therefore provided an opportunity to identify melanoma-associated APA by comparing immortalized melanocytes and melanoma cells in a consistent genetic background context.

We profiled global gene expression and poly(A) site selection in each of the two syngeneic cell lines with high-replicate RNA-seq and Poly(A)-seq (Fig. 2B). We identified significant differences in APA site selection between the two cell lines ($n = 6$ replicates per cell line) with APAlyzer[23]. This analysis revealed 654 significantly differentially polyadenylated transcripts for which differential APA site selection was supported by both Poly(A)-seq and RNA-seq (Fig. 2C–F; Supplementary Data File 3). 204 genes displayed strong correlations between APA and differential gene expression, but the majority did not, consistent with the idea that many APA events have gene- or protein-specific effects rather than simply modulating transcript levels in an easily predictable manner (Fig. 2G)[4].

We next compared APA in mouse and human melanoma. For each gene, we calculated the relative distal poly(A) site use in the B16-F10 mouse melanoma model system and for the human gene ortholog in the 424 TCGA melanoma samples and then computed the Pearson correlation for those gene-level values. This analysis revealed a modest but significant positive correlation between APA in mouse and human melanoma ($R = 0.24$, $p < 2.2 \times 10^{-16}$; Supplementary Fig. 3A–C). Overall, these data demonstrate that widespread alterations in APA characterize the B16-F10 model of mouse melanoma, suggesting that it may be a useful system for dissecting the functional contributions of individual APA events to tumorigenesis.

### CRISPR-Cas9 paired-guide RNAs enable functional manipulation of alternative polyadenylation

In order to identify possible functional roles for the APA events that we identified in melanoma, we sought to manipulate endogenous poly(A) site selection in order to test how each APA event influenced tumor growth. We utilized a well-established approach for endogenous poly(A) site manipulation, wherein CRISPR-Cas9 paired-guide RNAs (pgRNAs) are used to precisely delete an individual poly(A) site. This

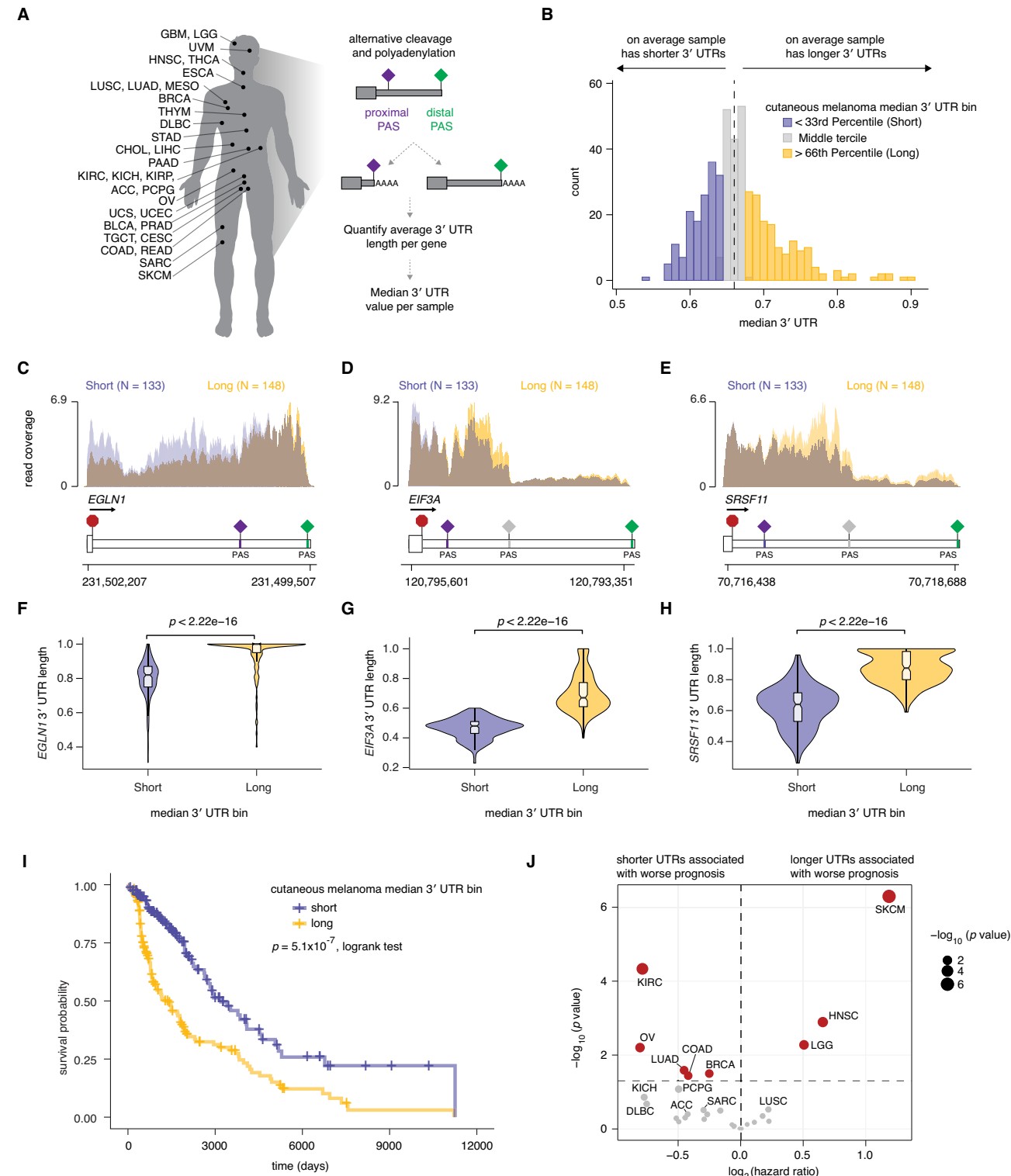

approach has been previously utilized by other groups to force usage of specific poly(A) sites in *TrpS3*, *Calm1*, and other genes[26–29].

We tested the efficacy of this approach in our melanoma model system using *Sap30l*, which has two distinct poly(A) sites separated by several kilobases. B16-F10 cells expressed SAP30L mRNA overwhelmingly utilizing the proximal poly(A) site, whereas Melan-A cells utilize the distal poly(A) site significantly more (Fig. 2C, F). We generated constitutive Cas9-expressing B16-F10 cells that we then transduced with two distinct pgRNAs designed to delete the proximal polyadenylation signal sequence within the *Sap30l* 3′ UTR (Fig. 3A, B).

Cas9-expressing B16-F10 cells treated with either pgRNA designed to knock out the proximal polyadenylation signal sequence of *Sap30l*, referred to as proximal knockouts (pKO), exhibited appreciable levels of genomic DNA excision at the expected site (Fig. 3C; Supplementary Fig. 4A, B). Poly(A)-seq of polyclonal cells treated with either of two distinct *Sap30l* pKO pgRNAs caused specific increases in use of the *Sap30l* distal poly(A) site (Fig. 3D; Supplementary Fig. 4C). We generated seven monoclonal lines with heterozygous excision of the proximal *Sap30l* polyadenylation signal, five monoclonal lines with homozygous excision, and six monoclonal negative control lines

**Fig. 1 | Global 3′ UTR length correlates with clinical outcomes across tumor types. A** TCGA cancer subtypes analyzed and schematic of quantification of gene-level 3′ UTR lengths for all genes per sample. We followed standard practice[11] to compute the median 3′ UTR length per sample, normalized such that a value closer to 0 indicates, on average, globally shorter 3′ UTRs and a value closer to 1 indicates globally longer 3′ UTRs. Values are based on the percent of distal poly(A) site usage index computed by the DaPars algorithm[15]. **B** Distribution of median 3′ UTR per sample for the 424 tumor samples in The Cancer Genome Atlas cutaneous melanoma cohort. Samples were binned into terciles corresponding to short (blue), medium (gray), and long (yellow) global 3′ UTR samples. The median of all samples is marked with a dashed black line. **C–E** Individual read coverage plots (read coverage reflective of TPM values) for three exemplar genes that exhibit significant differences in 3′ UTR length between samples with globally shorter (blue) and longer (yellow) UTRs as indicated in (**B**) for *EGLN1*, *EIF3A* and *SRSF11* in cutaneous melanoma (SKCM). Poly(A) signal sequences (PAS) sites indicated as diamonds. **F–H** Violin plots of gene level 3′ UTR length for *EGLN1*, *EIF3A* and *SRSF11* from (**C–E**) comparing short and long median 3′ UTR stratified cutaneous melanoma samples. A value closer to 0 indicates higher use of the proximal poly(A) site and a value closer to 1 indicates higher use of the distal poly(A) site (*p* values from two-sided Wilcoxon rank-sum test). Exact *P* values are $P = 1.178188^{-33}$, $2.047001^{-48}$ and $2.354266^{-40}$ for **F**, **G** and **H**, respectively. **I** Kaplan–Meier analysis comparing overall survival of TCGA cutaneous melanoma samples binned as short or long median 3′ UTR samples (*p* values from a two-sided logrank test). **J** Volcano plot of the log₂(hazard ratio) calculated from univariate cox regression models comparing overall survival of short versus long UTR stratified samples plotted against the −log₁₀(*p* value). *P* value from a Cox proportional hazard model. Cancer subtypes with a *p* value < 0.05 are indicated in red. Vertical dashed line indicates a log₂(hazard ratio) of 0, and the horizontal dashed line indicated a *P* value of 0.05.

transduced with a non-targeting control (NTC) pgRNA (Supplementary Fig. 4D). Treatment with either of the two distinct *Sap30l* pKO pgRNAs significantly increased use of the distal polyadenylation signal sequence in both the polyclonal and monoclonal settings as measured by RT-PCR, where cells with homozygous deletion of the proximal poly(A) signal displayed the highest use of the distal poly(A) site (Fig. 3E; Supplementary Fig. 4E). Using qRT-PCR to measure SAP30L mRNA levels, we found that homozygous excision of the proximal poly(A) signal led to a median reduction in total SAP30L mRNA levels of 47.9% (Fig. 3F; *p* = 0.017 computed across the monoclonal isolates). Consistent with this, we observed a strong negative correlation between distal polyadenylation site usage and total mRNA abundance for *Sap30l* (Fig. 3G; *p* = 0.0084). We concluded that targeted deletion of proximal poly(A) sites using a CRISPR-Cas9 pgRNA approach can effectively force distal poly(A) site use in both polyclonal and monoclonal populations of B16-F10 cells.

## High-throughput functional screening identifies alternative polyadenylation that regulates murine melanoma growth in vitro and in vivo

To test if APA is functionally relevant in melanoma, we utilized the CRISPR-Cas9-based approach to manipulating endogenous poly(A) site selection to create a multiplexed genetic screen. We designed a custom pgRNA library targeting 143 proximal polyadenylation signal sequences that we identified as cancer-associated from our genomic analyses (Supplementary Fig. 5A; Supplementary Data File 4). Approximately 25% of these APA events were dysregulated in both human and mouse melanoma (36 targets), while 75% were specific to mouse melanoma (107 targets) (Supplementary Fig. 5B). We created downloadable BED files to visualize pgRNA library design alongside the Poly(A)-seq data from B16-F10 and Melan-A cells that we used to select targets (Supplementary Fig. 6A, B). The final library comprises 8–10 unique targeting pgRNAs per proximal poly(A) signal, 150 positive control pgRNAs designed to knock out 15 distinct genes including core essential genes and tumor suppressor genes as negative and positive controls, respectively, and 150 negative control pgRNAs targeting proximal polyadenylation signals in genes that are not expressed in B16-F10 cells. We included ample negative control pgRNAs in order to allow for accurate determination of empirical false discovery rates (FDRs) and robust statistical significance testing (Supplementary Fig. 5B)[30]. The resulting targeting, positive control, and negative control pgRNAs each have similar distributions of on- and off-target scores relative to pgRNA libraries that we and others have previously used for functional interrogation of tumorigenesis (Supplementary Fig. 5C, D)[30,31]. We then cloned this library of 1718 unique pgRNAs via oligo array cloning as previously described[32] and confirmed excellent library diversity with next-generation sequencing of the final plasmid pool (Supplementary Fig. 5E, F).

We used this library to identify APA that influenced tumor growth by performing paired in vitro and in vivo screens in B16-F10 cells (Fig. 4A). In brief, we infected Cas9-expressing B16-F10 cells with the lentiviral library at an MOI of 0.2. 24 h later, we added 0.5 μg/mL puromycin, which we left for 2 days, and then cultured for one more day in puromycin-free media. We then split the cells into 8 replicates. For each replicate, we collected a day 0 fraction, plated 500,000 cells for culturing in vitro, or subcutaneously injected 500,000 cells into a mouse flank for tumor growth in vivo. After 20 days of monitored growth, cells and tumors were harvested, and total genomic DNA (gDNA) was extracted using established methods[33]. gDNA from the plasmid pool, day 0 time points, and day 20 in vitro *and* in vivo time points was then PCR amplified and subjected to next-generation sequencing with ~2500-fold coverage per pgRNA.

We performed enrichment analysis and FDR computation using our previously described statistical framework[30]. In brief, we normalized the fold-changes relative to day 0 for a given replicate such that the median of all pgRNAs targeting poly(A) sites in unexpressed genes was equal to 1 and pooled computed fold-changes across all replicates for a given condition to maximize statistical power (Supplementary Data File 5). We then calculated a *p* value for significant enrichment/ depletion associated with each poly(A) site target by comparing the distribution of fold-changes for all pgRNAs targeting that poly(A) site to the distribution of pgRNAs targeting poly(A) sites in unexpressed genes. We computed empirical FDRs for each targeted poly(A) site using a subsampling procedure across negative control pgRNAs targeting unexpressed genes (*n* = 10,000 samples).

Both positive and negative control pgRNAs performed as expected in vitro and in vivo (Fig. 4B–D; Supplementary Fig. 7A–C). pgRNAs designed to knock out genes that are essential for tumor growth exhibited marked depletion, while pgRNAs designed to delete proximal polyadenylation sites of unexpressed genes exhibited negligible enrichment or depletion, with 96.0% or 96.7% of pgRNAs falling within two standard deviations of the median fold-change for our in vitro and in vivo experiments, respectively (Fig. 4C, D; Supplementary Fig. 7A–E). In contrast, numerous pgRNAs targeting proximal polyadenylation sites of expressed genes exhibited significant enrichment or depletion, with deletion of some proximal polyadenylation sites associated with alterations in B16-F10 cell growth in vitro and in vivo of similar magnitudes to positive controls (Fig. 4C, D).

We next validated two particularly striking hits that emerged from our screen. pgRNAs targeting the proximal polyadenylation sites of *Atg7* and *Egln1* were strongly depleted and enriched, respectively, both in vitro and in vivo. We introduced individual pgRNAs targeting these polyadenylation sites into Cas9-expressing B16-F10 cells and systematically profiled polyadenylation site selection, protein levels, and effects on cell growth. All experiments were performed in comparison to Cas9-expressing B16-F10 cells transduced with a pgRNA targeting a poly(A) site in the unexpressed gene *Crabp1*, which was selected as a negative control because its normalized fold-change in the screen was close to the median of all control pgRNAs targeting poly(A) sites within unexpressed genes.

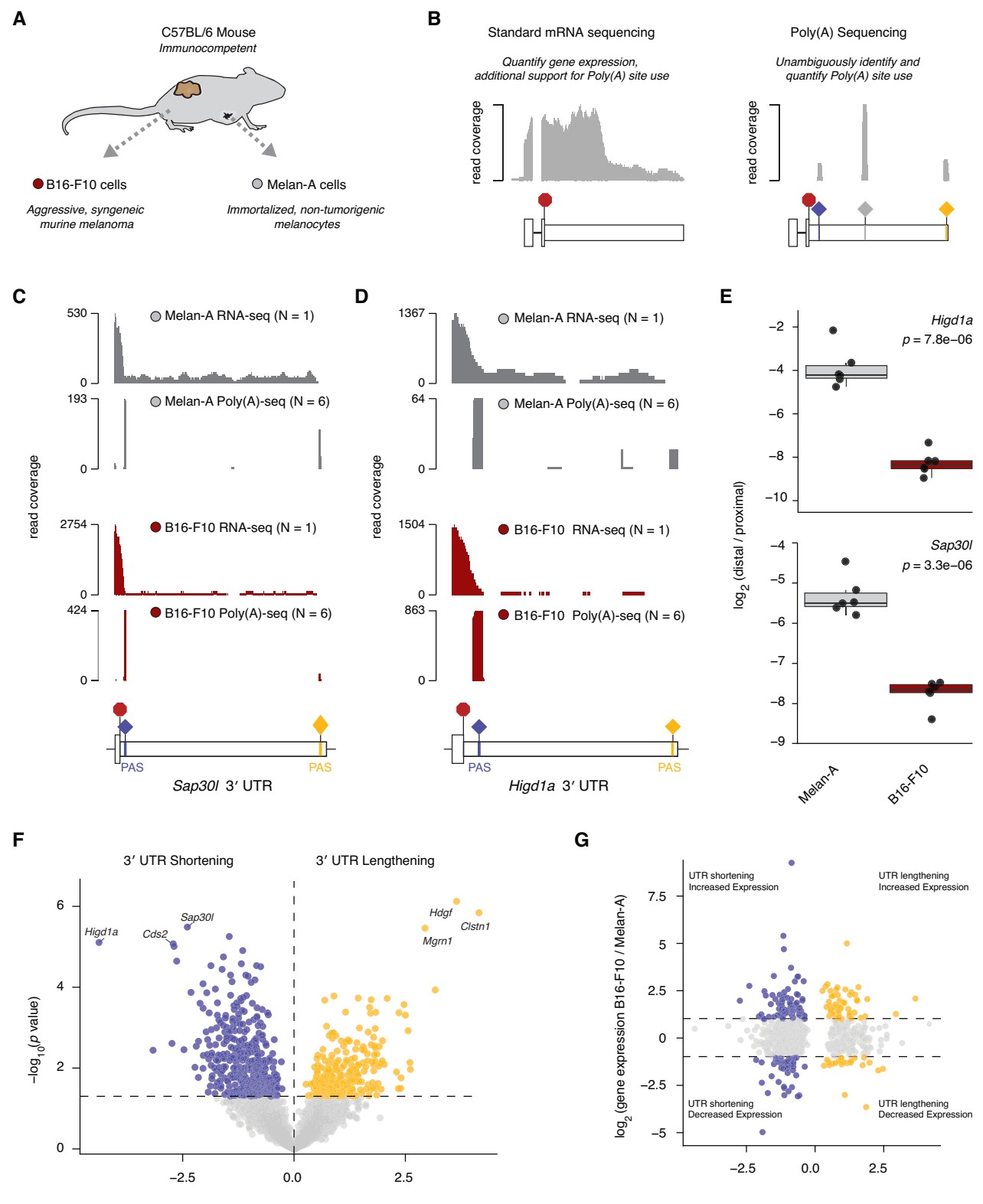

These experiments confirmed that poly(A) site knock out (pKO) pgRNAs targeting *Atg7* and *Egln1* induced significant increases in use of the distal polyadenylation sites, as intended (Fig. 4E, H). pKO pgRNA treatment significantly reduced ATG7 protein levels but was associated with only modest alterations in EGLN1 protein levels (Fig. 4F, I). Introduction of pKO pgRNAs to force use of distal polyadenylation sites in *Atg7* and *Egln1* recapitulated the in vitro growth phenotypes observed in the screen format (Fig. 4G, J). Overall, these

data demonstrate that multiplexed pgRNA screening effectively nominates specific APA events that influence melanoma cell growth both in vitro and in vivo.

### A long *Atg7* 3′ UTR suppresses murine melanoma growth in vitro and in vivo

We selected *Atg7* for additional studies due to its significant depletion both in vitro and in vivo. We performed a rescue experiment to

**Fig. 2 | Identification of differentially polyadenylated RNAs in a murine model of melanoma. A** Description of two syngeneic model cell lines, B16-F10 cells (red) and Melan-A cells (gray) both derived from a C57BL/6 background melanocyte origin. **B** Differences in information provided by RNA-seq and Poly(A)-seq methods, and examples of read coverage plots reflective of the reads generated by each sequencing approach across the terminal exon including the 3′ UTR of an example gene. **C** Read coverage plots of RNA-seq and Poly(A)-seq completed for Melan-A cells (gray) B16-F10 cells (red) of the *Sap30l* terminal exon and 3′ UTR with the stop codon and the annotated poly(A) signal sequences (PAS). **D** BAM coverage plot of RNA-seq and Poly(A)-seq completed for Melan-A cells (gray) and B16-F10 cells (red) of the *Higd1a* terminal exon and 3′ UTR with the stop codon and the annotated poly(A) signal sequences (PAS). **E** Box plot demonstrating relative 3′ UTR length calculated as the $\log_2$(distal reads / proximal reads) per sample for the gene *Higd1a* and *Sap30l*. Data reflects six Poly(A)-seq runs per cell line. *P* values from two-sided Wilcoxon rank-sum test. **F** Volcano plot of all differentially polyadenylated

transcripts between B16-F10 cells and Melan-A cells quantified using the APAlyzer pipeline. Data reflects six Poly(A)-seq runs per cell line, significantly altered events were determined using a two-sided Student's *t* test. Significantly shortened 3′ UTRs are indicated in blue and significantly lengthened 3′ UTRs are indicated in yellow. Vertical dashed line indicates a difference of 0, and the horizontal dashed line indicated a *P* value of 0.05. **G** Scatter plot of all genes identified as differentially polyadenylated from Poly(A)-seq data, comparing gene-level 3′ UTR length differences and gene expression differences between B16-F10 and Melan-A cells. Blue indicates genes that are significantly shortened in B16-F10 cells and display a significant difference in expression levels, yellow indicates genes that are significantly lengthened and display a significant difference in expression levels, and gray indicates the gene shows no significant change in expression between the two cell lines. The horizontal dashed lines indicate a $\log_2$(fold-change) in gene expression of 1 and −1.

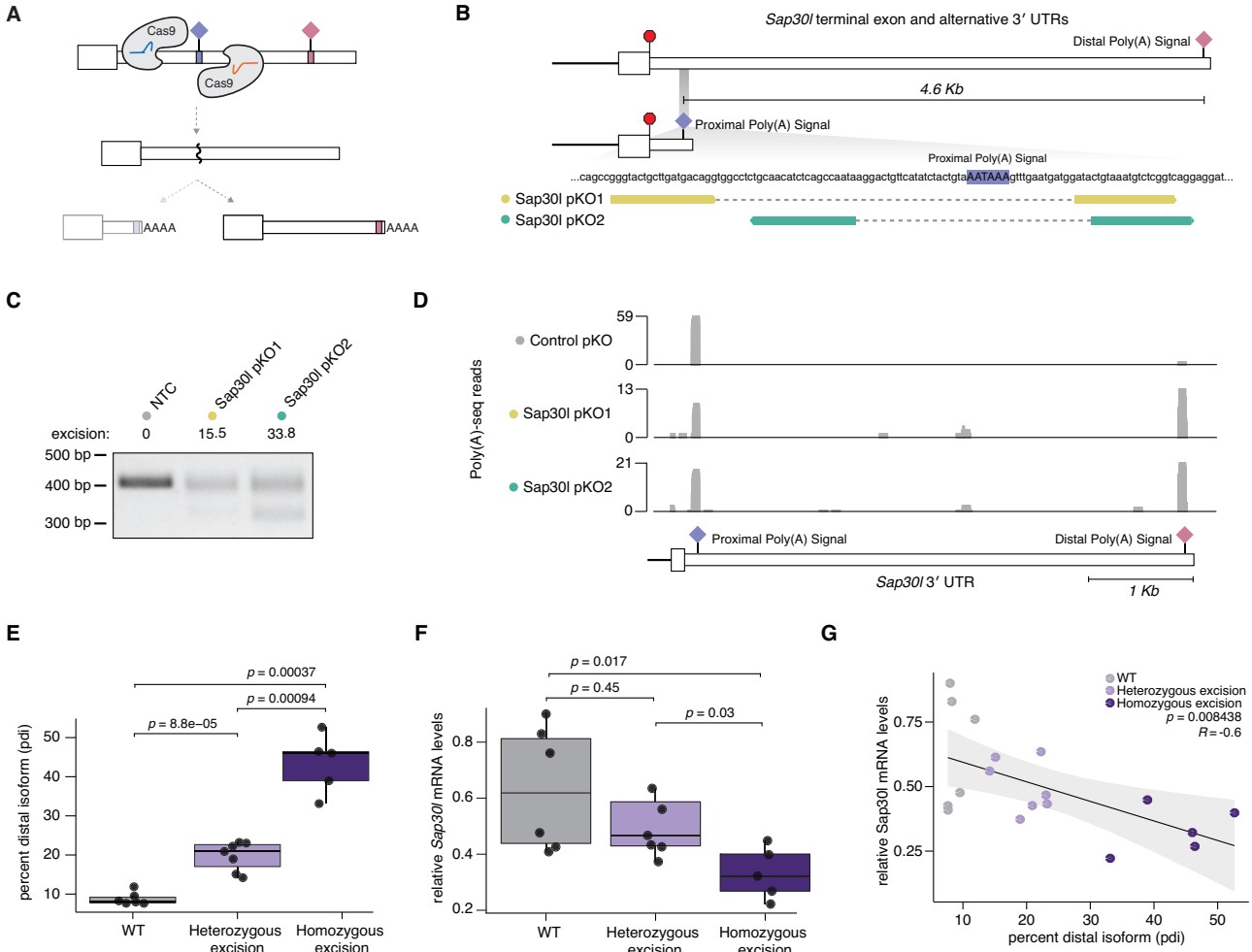

**Fig. 3 | A CRISPR-Cas9 paired-guide RNA strategy can force targeted distal poly(A) site usage. A** Schematic of CRISPR-Cas9 paired-guide RNA (pgRNA) approach to excise proximal poly(A) sites and force use of distal poly(A) sites. **B** Diagram of *Sap30l* terminal exon with two distinct 3′ UTRs which vary depending on use of a proximal or distal poly(A) signal. Schematic of two distinct sets of proximal poly(A) KO (pKO) pgRNAs designed to delete the proximal poly(A) signal in *Sap30l*, annotated as *Sap30l* pKO1 and pKO2. **C** Genotyping PCR of polyclonal B16-F10 Cas9-expressing cells treated with a non-targeting control (NTC) or one of two distinct *Sap30l* pKO pgRNAs, with the calculated percent of signal showing DNA excision (representative gel from *n* = 3 biological replicates). **D** Read coverage plots of Poly(A)-seq completed for Cas9-expressing B16-F10 cells treated with either a control pgRNA (gray), *Sap30l* pKO1 (green), or *Sap30l* pKO1 (blue). Schematic illustrates the *Sap30l* terminal exon and 3′ UTR with the annotated poly(A) signal sequences (PAS). **E** Box plots of the percent distal isoform (pdi) usage per cell

line quantified from the nested RT-PCR of the *Sap30l* 3′ UTR for each monoclonal cell line grouped by genotype (WT, heterozygous or homozygous deletion of the proximal poly(A) signal). Data reflects six, seven, and five distinct monoclonal lines with wild-type, heterozygous, or homozygous deletion of the proximal poly(A) signal sequence, respectively. *P* value from a two-sided Wilcoxon rank-sum test. **F** Box plots of *Sap30l* mRNA abundance per cell line measured by q-RT-PCR for each monoclonal cell line grouped by genotype (WT, heterozygous or homozygous deletion of the proximal poly(A) signal). Data reflects six, seven, and five distinct monoclonal lines with wild-type, heterozygous, or homozygous deletion of the proximal poly(A) signal sequence, respectively. *P* value from a two-sided Wilcoxon rank-sum test. **G** Scatter plot and Pearson correlation of *Sap30l* percent distal isoform usage versus *Sap30l* mRNA abundance. Gray error band represents 95% confidence interval. *P* value from a Pearson correlation.

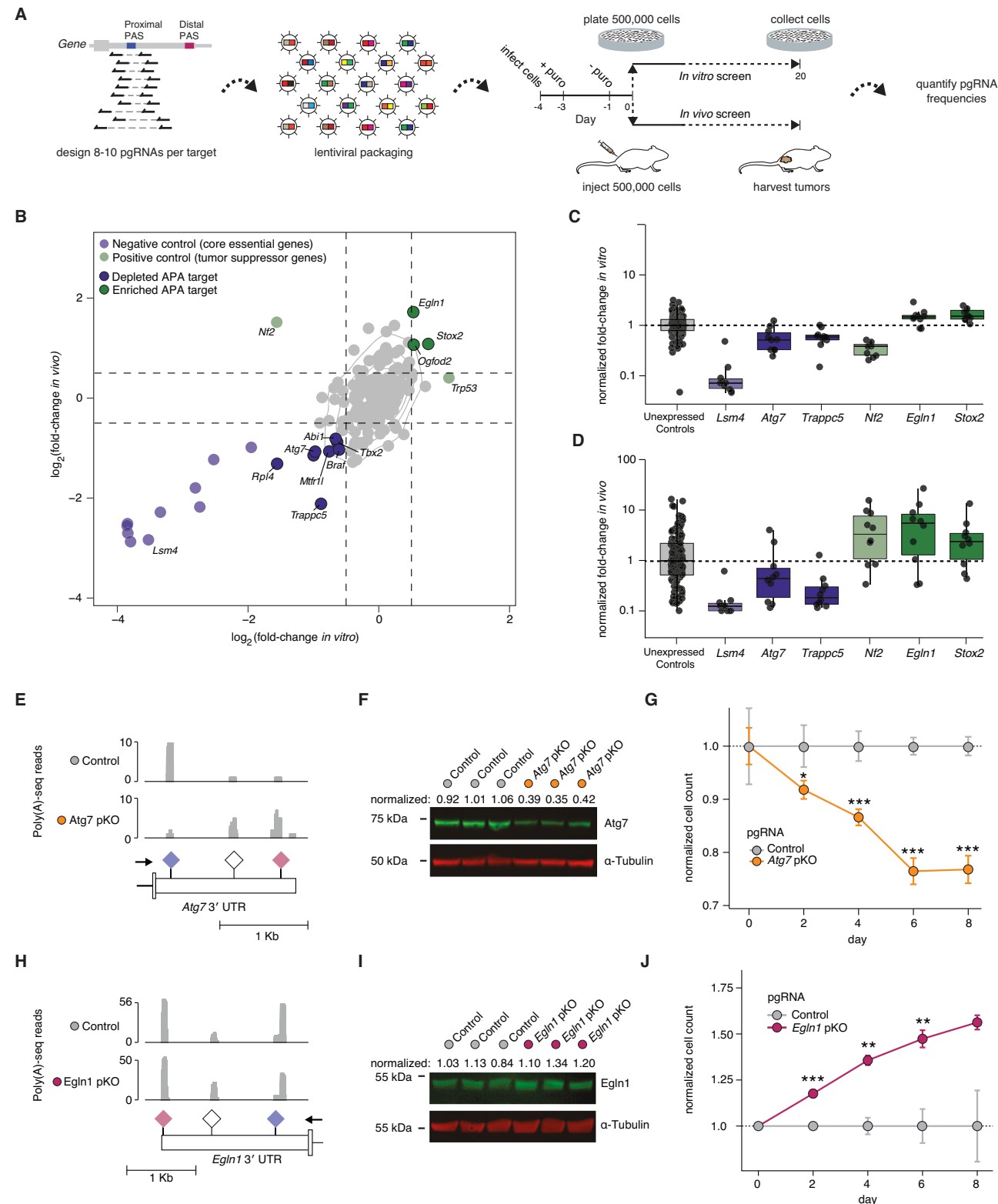

confirm that reduced levels of the short *Atg7* isoform were responsible for the reduced cellular growth that we observed in our *Atg7* pKO B16-F10 cells (Fig. 4G) by expressing transgenic *Atg7* mRNA in the *Atg7* pKO background and measuring cell growth. We generated a cDNA construct harboring the *Atg7* coding sequence and the endogenous short *Atg7* 3′ UTR sequence as well as 200 nucleotides downstream of the poly(A) signal sequence (Supplementary Fig. 8A). We additionally generated a construct with the *Atg7* coding sequence and an SV40

poly(A) signal. Stable integration and expression of either construct in the *Atg7* pKO background restored ATG7 protein levels and rescued the cellular growth phenotype observed in *Atg7* pKO cells (Supplementary Fig. 8B, C). We observed rescue of cell growth for both constructs (with the endogenous short *Atg7* 3′ UTR and with the SV40 poly(A) signal), suggesting that decreased ATG7 protein levels, rather than low levels of the short *Atg7* 3′ UTR-containing mRNA itself, underlies the reduced growth characteristic of *Atg7* pKO cells.

**Fig. 4 | A high-throughput functional CRISPR-Cas9 screen reveals APA events that influence melanoma growth. A** Schematic of CRISPR-Cas9 pgRNA library design (8–10 pgRNAs per target), cloning and paired in vitro and in vivo screen format, and timing. **B** Mean $\log_2$ (fold-change) of all pgRNAs targeting a specific gene or proximal poly(A) site normalized to negative control pgRNAs targeting proximal poly(A) sites of unexpressed genes for paired in vitro and in vivo screens in Cas9-expressing B16-F10 cells ($n = 8$ biological replicates). Black outlines, significantly enriched (green) or depleted (purple) proximal poly(A) targets. No outlines, positive (light green) and negative (light purple) control genes. Dashed lines indicate a $\log_2$(fold-change) of 0.5 and −0.5, respectively for in vitro (vertical) and in vivo (horizontal) data. **C** Log fold-changes associated with the indicated targets. Each point, reflects the mean value of a single pgRNA across $n = 8$ biological replicates. Dashed line indicates a normalized fold-change of 1. **D** As (C), but in vivo. Dashed line indicates a normalized fold-change of 1. **E** Read coverage plots of Poly(A)-seq data for Cas9-expressing B16-F10 cells treated with either a control pgRNA (gray) or *Atg7* pKO1 (orange). Schematic shows the *Atg7* terminal exon and 3′ UTR with the annotated poly(A) signal sequences (PAS). **F** Western blot of lysates from Cas9-expressing B16-F10 cells treated with a control pgRNA or *Atg7* pKO pgRNA. ATG7 protein level normalized to alpha-tubulin control. **G** In vitro cell growth of Cas9-expressing B16-F10 cells treated with a control pgRNA or *Atg7* pKO pgRNA measured by CellTiter-Glo. Measurement is average of three experimental replicates +/- S.E.M. Significance denoted as $*p < 0.05$, $**p < 0.01$ or $***p < 0.001$ using a two-sided Student's $t$ test (exact $p$ values 0.023, 0.00011, $9 \times 10^{-8}$, and 0.00096). **H** As (**E**), but for *Egln1* and *Egln1* pKO1. **I** Western blot of lysates from Cas9-expressing B16-F10 cells treated with a control pgRNA or *Egln1* pKO pgRNA. Egln1 protein level normalized to alpha-tubulin control. **J** In vitro cell growth of Cas9-expressing B16-F10 cells treated with a control pgRNA or *Egln1* pKO pgRNA measured by CellTiter-Glo. Measurement is the average of three experimental replicates +/- S.E.M. Significance denoted as $*p < 0.05$, $**p < 0.01$ or $***p < 0.001$ using a two-sided Student's $t$ test (exact $p$ values 0.0005, 0.0011, 0.0079, and 0.059).

To further characterize differences between the *Atg7* short and long 3′ UTR isoforms, we completed an Actinomycin D time course to assess mRNA stability of the respective mRNAs. We found that the percent of all *Atg7* mRNA utilizing the long isoform decreased relative to the fraction of total *Atg7* mRNA across the Actinomycin D treatment time course (Fig. 5A, B). These data demonstrate that the *Atg7* mRNA utilizing the long 3′ UTR is less stable than the *Atg7* mRNA utilizing the short 3′ UTR, likely explaining at least in part the reduced protein levels we observed in B16-F10 *Atg7* pKO cells (Fig. 4F).

We then sought to replicate the in vivo growth phenotype observed in the CRISPR screen for *Atg7* pKO cells in focused assays. Cas9-expressing B16-F10 cells were treated with either a pKO pgRNA targeting *Atg7* or a control pgRNA targeting a poly(A) site in an unexpressed gene, and then engrafted into C57BL/6 mice. Consistent with results from the multiplexed screen, *Atg7* pKO tumors exhibited reduced growth in vivo, leading to significantly prolonged host survival using Kaplan–Meier survival analysis ($p = 0.0053$) (Fig. 5C; Supplementary Fig. 9A, B).

We next sought to understand the molecular consequences of *Atg7* APA. Autophagy related protein 7, ATG7, is considered a critical protein for autophagosome formation and function, and loss of ATG7 protein inhibits autophagy[34]. We therefore investigated the consequences of *Atg7* APA for autophagy. Complete *Atg7* knock out and consequent loss of ATG7 protein caused a significant increase in p62 protein levels, as expected (Supplementary Fig. 10A). In contrast, treating cells with an *Atg7* pKO pgRNA to force distal polyadenylation site usage did not cause detectable deficits in autophagy at baseline or following exposure to serum and amino acid starvation, measured by both p62 accumulation and a flow cytometry-based LC3-GFP-mCherry reporter (Supplementary Fig. 10A–E)[35]. These results are distinct from a previous study reporting that *ATG7* 3′ UTR lengthening was associated with autophagy inhibition in pro-B cells, a difference that may be attributable to cell type-specific effects or that study's reliance on KD of all ATG7 isoforms for functional studies rather than specific manipulation of poly(A) site usage[36]. Although forcing distal polyadenylation site usage reduced ATG7 protein levels by ~60% (Fig. 4F; Supplementary Fig. 10A), we detected none of the autophagy impairment that was readily apparent upon *Atg7* KO, highlighting the utility of directly manipulating poly(A) site usage for functional inference.

We next performed immunohistochemistry (IHC) for the Ki67 proliferation marker on control and *Atg7* pKO tumors. Although *Atg7* pKO tumors grew slower (Fig. 5C; Supplementary Fig. 9A, B), they exhibited a significantly higher fraction of nuclei with moderate and strong Ki67 staining relative to control tumors (Fig. 5D, E; Supplementary Fig. 9C, D). The levels of Ki67 fluctuate significantly throughout the cell cycle, with RNA and protein levels rising through S phase and peaking in G2/M phase[37]. We therefore performed cell cycle analysis with propidium iodide staining and found that *Atg7* pKO increased the fraction of B16-F10 cells in S phase and G2/M phase. These altered cell cycle kinetics are consistent with the increased Ki67 staining intensity observed by tumor IHC and concordant with a previous study demonstrating that *ATG7* KD by siRNA in human bladder cancer cells increased the fraction of cells in S phase and G2/M phase (Fig. 5F; Supplementary Fig. 9E, F)[38].

*Atg7* undergoes APA in both mouse and human cells, raising the possibility that *ATG7* 3′ UTR length could influence human melanoma growth just as it does mouse melanoma. We first assessed mutation status of *ATG7* across the 424 patients in TCGA's melanoma cohort. No nonsense, frame-disrupting, or recurrent missense mutations were observed across this cohort, confirming that *ATG7* is not a common mutational target in clinical melanoma (Fig. 5G). We then performed Kaplan–Meier survival analysis after stratifying these 424 cutaneous melanoma patients by *ATG7* 3′ UTR length. We observed that patients whose tumors expressed a longer *ATG7* 3′ UTR exhibited significantly better progression-free survival, consistent with our finding that forcing long *Atg7* 3′ UTR usage slowed mouse melanoma growth (Fig. 5H). Given this strong association between *ATG7* APA and survival that we observed in both mouse and clinical human cancer, it is interesting to speculate that *ATG7* APA provides a post-transcriptional mechanism by which cancers may regulate ATG7 protein levels to influence their growth even in the absence of *ATG7* mutations.

## Discussion

Dysregulated alternative polyadenylation is a pervasive feature of most cancers, and thousands of cancer-associated APA events have been identified. However, very few of these APA events have been functionally studied or linked to cancer phenotypes. Here, we describe a high-throughput, in vivo screening platform to interrogate the functional roles of individual APA events in cancer and identify APA isoforms which enhance or reduce tumor growth in an immuno-competent mouse model. Our study systematically reveals individual APA events that play a causative, rather than simply a correlative, role in influencing key cancer phenotypes.

We primarily focused on APA events whose modulation was associated with notable effects on tumor growth. However, it is interesting to note that most APA events queried in our screen were not associated with detectable alterations in tumor cell growth either in vitro or in vivo. Although we cannot rule out the possibility that such negative results arise from ineffective pgRNA targeting and APA modulation, this seems unlikely given the relatively high efficiency of APA modulation that we observed even in polyclonal settings for all APA events that we studied individually. These data therefore suggest that many APA events have modest or no effects on tumor growth

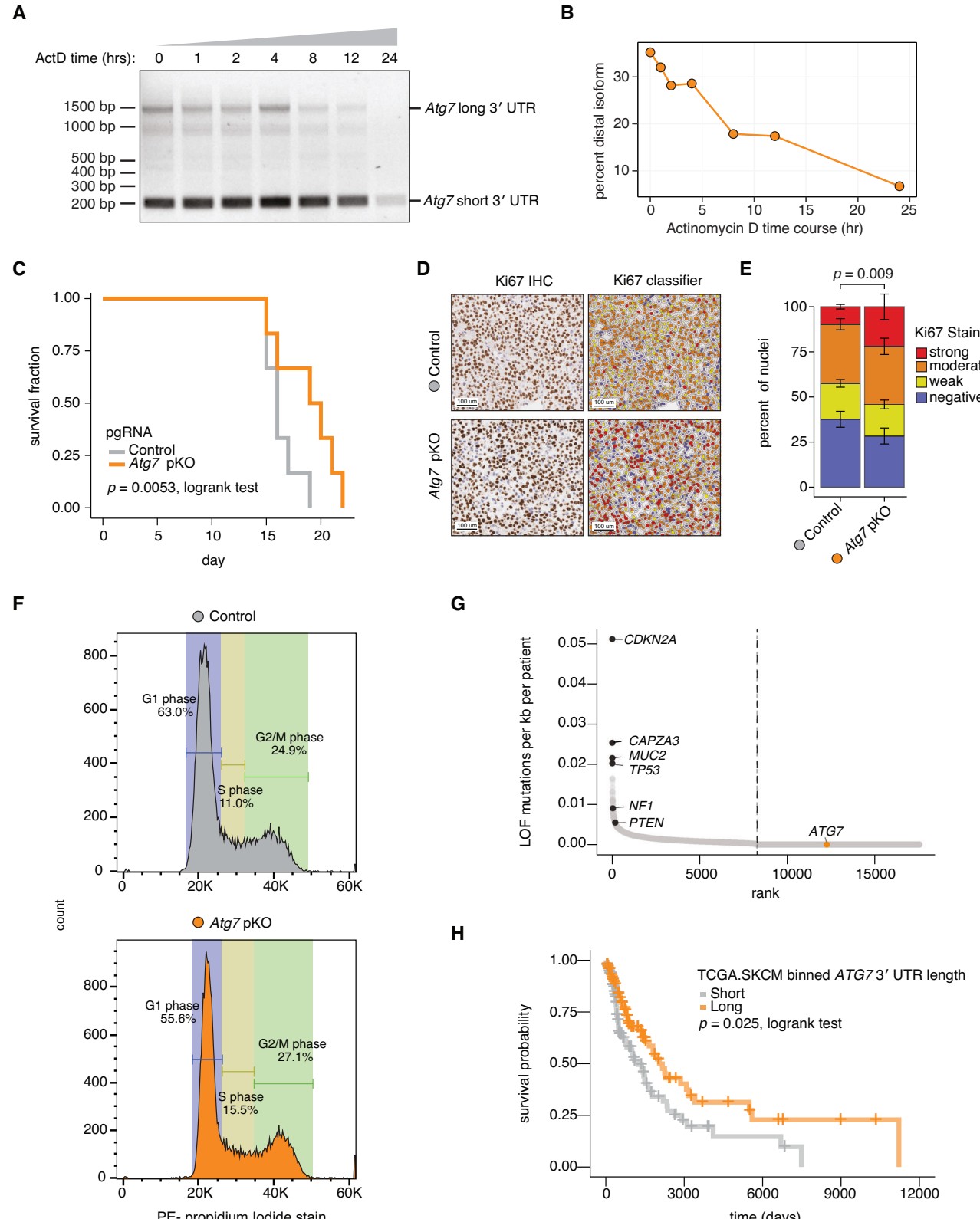

kinetics in our system and therefore either alter other facets of tumorigenesis or are bystander events of low functional relevance. This dichotomy, where some APA events play critical functional roles and others do not, may help to resolve the ongoing controversy regarding the likely functional relevance of most APA[12,14]. In either case, our data highlight the utility of high-throughput screens for identifying functionally relevant events for detailed study.

It is important to note that our in vivo studies focused on determining of how APA modulation altered tumor cell growth, which is only one of many important cancer phenotypes that influence patient prognosis. For example, overall survival is frequently strongly associated with metastasis, which our assays did not measure. A natural and important extension of the current study would be to determine how cancer-specific APA influenced tumor cell dissemination.

**Fig. 5 | *Atg7* alternative poly(A) site selection alters melanoma cell growth in vitro and in vivo. A** Nested RT-PCR of the *Atg7* 3′ UTR in Cas9-expressing B16-F10 cell lines treated with 1 µg/mL Actinomycin D to inhibit transcription. Expected *Atg7* 3′ UTR size when utilizing the proximal poly(A) site (short) or the distal poly(A) site (long) are indicated. Representative gel from *n* = 2 biological replicates. **B** Quantification of the percent distal isoform from (A). Gel intensities quantified using FIJI. **C** Survival data from a cohort of C57BL/6 mice injected with Cas9-expressing B16-F10 cells treated with a control pgRNA targeting the poly(A) site of an unexpressed gene (control) or an *Atg7* pKO pgRNA (*n* = 6 mice, 12 tumors, per condition). **D** Left, representative immunohistochemistry images of control or *Atg7* pKO pgRNA tumor sections stained for Ki67. Right, images with nuclei classified using HALO as negative (blue), weak (yellow), moderate (orange) or strong (red) staining. **E** Stacked bar plot quantifying data from (D). *n* = 4 images from 4 distinct

tumors per genotype. For each image, the entire slide is processed, only excluding areas if they are easily discernible as non-tumor tissue. *P* value calculated with a two-sided binomial proportion test. **F** Representative flow cytometry histograms of Cas9-expressing B16-F10 cells treated with a control or *Atg7* pKO pgRNA cells stained with propidium iodide and then analyzed using Dean-Jett-Fox classification for cell cycle stage from FlowJo v10. **G** Rank order plot of loss-of-function (LOF) mutations normalized per kilobase per patient detected in genomic DNA sequencing from TCGA cutaneous melanoma cohort (*n* = 424 patients). Vertical dashed line, point at which all remaining ranked genes have no detected LOF mutations. **H** Kaplan–Meier analysis of progression-free survival in TCGA cutaneous melanoma cohort for patients binned into terciles based on *ATG7* 3′ UTR length. *P* value from a two-sided logrank test.

We utilized the CRISPR-Cas9 pgRNA system for endogenous APA modulation because it has been extensively tested and validated[26–29]. Future studies of APA may take advantage of the diverse other genome editing technologies that are now amenable to high-throughput screens. For example, dead Cas9 (dCas9) targeted downstream of the poly(A) signal has been used to enhance proximal poly(A) site selection[39], providing a useful counterpart to the suppression of poly(A) site use via pgRNAs that we relied upon. A fusion of the RNA-targeting dead Cas13b with adenosine deaminase 2 (ADAR2) can specifically modify adenosine residues in RNA[40] and as such could potentially be used to disrupt the A-rich canonical polyadenylation signal sequences AATAAA and ATTAAA, thereby allowing for functional manipulation of poly(A) sites at the level of RNA, rather than DNA.

Although we focused on cancer in the current study, we speculate that high-throughput functional interrogation of APA will prove equally fruitful in other biological contexts. For example, embryonic development and tissue differentiation[41,42] display coordinated, temporal regulation of poly(A) site selection, while poly(A) site selection is globally dysregulated in many genetic diseases, including Oculopharyngeal Muscular Dystrophy and Duchenne Muscular Dystrophy[43,44]. Targeted sequencing has cataloged thousands of context-specific APA alterations in these and other systems, the vast majority of which have never been functionally studied. High-throughput manipulation of endogenous poly(A) site selection offers an efficient way to bridge the gap between APA identification and functional understanding.

## Methods

### Animal use
All animal work and procedures were completed in accordance with the Guidelines for the Care and Use of Laboratory Animals and approved by the Fred Hutchinson Cancer Center Institutional Animal Care and Use Committee. Six-week-old C57BL/6 male mice were obtained from the Jackson Laboratory for all animal experiments listed here. Animals were housed for 1 week prior to initiation of any experiments in line with IACUC guidance.

### Cell lines and culture
B16-F10 cells were obtained from ATCC (CRL-6475) and cultured per the manufacturer's instructions. Cell lines tested negative for mycoplasma contamination. Cell lines were authenticated via RNA-seq expression profiling. B16-F10 cells were also spot checked as they express melanin and cell pellets are black (no other cell lines in use are black when pelleted). The Melan-A cell line was obtained from the Wellcome Trust Functional Genomics Cell Bank and cultured per their instructions[24]. B16-F10 cells were infected at 0.3 MOI with Cas9 lentivirus (Addgene 52962-LV) and selected with 2 ug/mL Blasticidin for 7 days. Polyclonal cells were then single cell sorted, and individual clones were assessed for on-target editing efficiency, all subsequent CRISPR assays were completed with this B16-F10 Cas9 clone with the highest editing rate.

### Poly(A)-Seq library preparation
RNA was isolated from cell pellets with the Direct-zol RNA MiniPrep kit (Zymo Research). Protocol was adapted from Derti et al.[1]. In brief, 5 µg of RNA was heated at 65° for 2 min and then placed on ice. Poly(A) selected mRNA was isolated using DynaBeads (ThermoFisher 61006) per the manufacturer's instructions. 1st strand cDNA synthesis using SuperScript IV Reverse Transcriptase (ThermoFisher 18090010) and the primer RKB4087 (a modified oligo d(TVN) primer) to generate cDNA reads which begin synthesis just before the poly(A) tail. This mixture was then purified using a 2x SPRI bead clean up and then followed by 2nd strand synthesis and then a 1x SPRI bead clean up. This cDNA library was then subject to PCR (NEBNext High Fidelity 2x Master Mix) and primers RKB4089 (universal primer) and RKB4090-RKB4101 (indexed primers). Specific primer sequences are listed in Supplementary Data 6. The final libraries generate a broad range of fragment sizes, as such we gel extracted libraries from a 2% agarose gel corresponding to 300–500 bp size. Following extraction and purification, library size was analyzed with a 4200 TapeStation System before sequencing.

### RNA-seq library preparation
RNA was isolated from cell pellets with the Direct-zol RNA MiniPrep kit (Zymo Research). Poly(A)-selected, unstranded Illumina libraries were prepared following the TruSeq protocol per the manufacturer's instructions. Library size and distribution was analyzed with a 4200 TapeStation System before sequencing on an Illumina HiSeq as 2 × 50 bp to obtain ~40 million reads per sample.

### RNA-seq and Poly(A)-seq data analysis
RNA-seq was analyzed as previously described[45]. RNA-seq reads were mapped to an annotated transcriptome created using Ensembl 71[45,46], UCSC knownGene[47] and Misov2.0[48] annotations using RSEM version 1.2.4[49] (modified to call Bowtie[50] with option '-v 2'). Unaligned reads were then mapped to the corresponding genome (hg19/GRCh37 assembly, mm10/GRCmc38 assembly) and a database containing all possible pairings of 5′ and 3′ splice sites per gene in our merged transcriptome annotation using TopHat version 20.8b[51]. Mapped reads were then merged and input into MISO v2.0. For TCGA studies, we analyzed 9045 available samples across 29 cancer types.

For Poly(A)-seq, data was mapped similarly but in a stranded fashion. BAM files were input into the APAlyzer package in R[23] and the gene level $\log_2$(distal reads/proximal reads) was computed per sample for each respective strand.

### Survival analyses
Cancer type abbreviations are the same as TCGA standards (https://gdc.cancer.gov/resources-tcga-users/tcga-code-tables/tcga-study-abbreviations). Survival analyses were completed with the Kaplan–Meier estimator and statistical test were performed with a logrank test (R package survival). Stratification per cancer subtype was completed for Fig. 1b, Supplementary Figs. 1A–C and 2A–F by

computing the median 3′ UTR length per sample and dividing each cancer subtype into terciles (short, medium and long) and comparing the short (≤33%) vs long (≥66%) bins. Gene level 3′ UTR measurements were downloaded from a previously published database[52] or computed using the APAlyzer package in R[23].

For Fig. 5F, patients from the TCGA SKCM cohort were stratified based on the ATG7 3 UTR length per sample, again into terciles and downstream analyses were performed identically as described above.

## pgRNA library construction and cloning

Target genes selected from the Poly(A) sequencing and TCGA SKCM analysis were largely selected based on manual inspection of BAM coverage plot quality. For each gene, the most utilized APA sites within the 3′ UTR were identified from PolyA-DB, filtered based off number of tissues where a particular APA site is utilized and the fraction of samples it has been detected in[23]. Of those Poly(A) sites, the most proximal site was selected for targeting. The 100 base-pairs up and downstream of the given poly(A) site were selected as the genomic window wherein all possible gRNAs were generated using Guidescan Version 1.0[53]. The downloaded gRNAs were then identified as up or downstream of the poly(A) signal (PAS), and then the guides were combined in a pairwise fashion to generate all possible pgRNAs which disrupt the poly(A) signal sequence. For each target we excluded gRNAs with more than two two-nucleotide off-target sites, more than thirty three-nucleotide off-target sites or an on-target editing efficiency of less than thirty according to Rule Set 2 scoring[54]. All input target sites were then filtered to only include targets with at least eight pgRNAs. For sites with more than ten possible pgRNAs, pgRNAs were ranked based first on targeting efficiency and then specificity to select the ten pgRNAs to be utilized in the library. This same process was used to generate all possible pgRNAs targeting highly utilized PAS sites of genes that are not expressed in B16-F10 or Melan-A cells. In this control library, 150 pgRNAs were selected randomly as controls to be included in the library, of note this random selection process was run several times until the distribution of control pgRNAs matched the true library in terms of efficiency and specificity. An additional 15 genes were included as growth controls based on previous literature, either core essential genes identified from DepMap or genes which promote more rapid B16-F10 cell growth in vivo identified by mining data generated from previous genome wide gene knock out CRISPR screens in B16-F10 cells[55].

## gDNA PCR and on-target editing verification

gDNA was extracted using the DNeasy Blood and Tissue kit (Qiagen) per the manufacturer's protocol. A window around the *Sap30l* proximal poly(A) site was selected to allow for simple detection of deletion by band separation on gel electrophoresis. PCR amplicons were then purified and submitted for AmpliconEZ sequencing (Azenta/Genewiz). Reads were trimmed, mapped and the fraction of reads with an intact or disrupted poly(A) signal sequence were identified using the CRISPresso2 software[56,57]. Specific primer sequences are listed in Supplementary Data 6.

## pgRNA screen

$9 \times 10^7$ Cas9 expressing B16-F10 cells were infected at an MOI of 0.2. Twenty-four hours post infection, cells were then selected with 1 μg/mL Puromycin for 48 h. Remaining cells were then pooled and divided into eight replicates. 500,000 cells were isolated from each replicated and frozen as a time point 0. Of the remaining cells, 500,000 cells were then plated into 10 cm plates and another 500,000 cells were injected subcutaneously into C57BL/6 mice. Cells grown in culture were passed every 3–4 days, and tumors were monitored and then harvested when reaching 1.5 cm in any single dimension or after 20 days, whichever condition was met first. On the day a given tumor was harvested, digested and frozen, the corresponding cells growing in vitro were also frozen. gDNA was then extracted from all samples using previously published methods[33], and quantified via Nanodrop. 1.5 μg of each sample was then utilized as input for PCR to amplify out integrated pgRNA constructs as previously described[30,31]. Final, purified libraries were then submitted for Next-Generation Sequencing using a custom approach performed as previously described.

## pgRNA sequencing and data analysis

Analyses were performed identically as previously described[30]. In brief, each read includes independent reads for gRNA1 and gRNA2 as well as an index read. Each gRNA was separately mapped to a database of pgRNAs using BowTie, and correct pairings were identified by ensuring gRNA1 and gRNA2 come from the same pgRNA, incorrect pairings were discarded. We generated a per pgRNA pseudocount which was added to the raw counts and served to regularize the fold-change computations so that they are proportional to the relative representation of each individual pgRNA in the library.

Fold-changes were then computed by comparing each time point to the counts at the day 0 time point. These fold-changes were then normalized so that the median of all pgRNAs targeting poly(A) sites in unexpressed genes (150 pgRNAs in total) was equal to 1. For each poly(A) site target or gene KO control target a *p* value was computed by comparing the distribution of normalized $\log_2$ fold-changes per pgRNA (8–10 per target) to the distribution of all 150 pgRNAs targeting poly(A) sites in unexpressed genes using a two-sided Wilcoxon rank-sum test. FDRs were computed by generating a distribution of fake target *p* values by randomly subsampling 10 pgRNAs from our control pgRNAs 10,000 times to estimate the distribution of expected *p* values. We then estimated FDR per real poly(A) site or gene KO target via the cumulative distribution function of the distribution of the fake target *p* values.

## In vitro validation studies

Cas9 expressing B16-F10 cells were grown in standard conditions and then transduced with lentivirus containing indicated pgRNAs and then selected in 1 μg/mL Puromycin for 72 h. For cell growth assays, 100,000 cells were plated into each well of a 24 well plate and 20% of the well was passaged every 2 days into a new 24 well plate. 100 μL of the remaining cell suspension was used as input for a CellTiter-Glo (ProMega Catalog Number G9242) assay performed per the manufacturer's instructions.

## In vivo validation studies

Cas9 expressing B16-F10 cells were grown in standard conditions and then transduced with lentivirus containing indicated pgRNAs and then selected in 1 μg/mL Puromycin for 72 h. $5 \times 10^6$ cells were then injected subcutaneously into each flank of adult male C57BL/6 mice, and monitored using calipers. Animals were euthanized when a tumor reached 1.5 cm in any dimension in accordance with IACUC guidelines, and no tumors ever exceeded this threshold in any dimension. Tumor material was isolated either for an archived flash frozen sample, fixed in 10% formalin at room temperature for histology studies or placed in TRIzol reagent (ThermoFisher 15596018) for subsequent RNA isolation.

## Immunohistochemistry

Tissues from tumors were processed, embedded and stained through the Fred Hutch Experimental Histopathology core. Mouse Ki-67 (CST Clone 12202 1:2000 dilution) staining was performed using a rabbit monoclonal antibody. Staining was performed with a BOND RX autostainer (Leica Biosystems) and images were then acquired with an Aperio ImageScope at 40x magnification (Leica Biosystems). Image analysis was completed using HALO Image Analysis software.

## RT-PCR analyses

RNA was isolated using the Direct-zol RNA MiniPrep (Zymo Research). SuperScript IV reverse Transcriptase was used to synthesize cDNA per the manufacturer's instructions, but using a specific oligo d(TVN) primer to amplify the DNA directly upstream of the poly(A) tail (ThermoFisher Scientific). Nested RT-PCR was performed with a universal poly(A) tail primer and two gene specific primers to amplify the 3′ UTR. The final product visualized using agarose gel electrophoresis and band intensity quantification was performed with FIJI/ImageJ and reported as the percent distal isoform (pdi) defined as the percent of total signal arising from bands corresponding to distal poly(A) site use. For the Actinomycin D treatment time course, cells were plated and treated with 1 μg/mL Actinomycin D (Sigma-Aldrich A1410-2MG) for the indicated time frame, whereafter cells were harvested and RNA was isolated as described above. Specific primer sequences are listed in Supplementary Data 6.

## Western blotting

Total protein lysates were isolated in 1x RIPA buffer and quantified with the Pierce 660 nm Protein Assay Reagent. Total protein lysates were electrophoretically separated and then transferred onto a nitrocellulose membrane using the NuPAGE system (ThermoFisher Scientific). Each membrane was blocked for 1 h at room temperature and then probed with primary antibody diluted in a blocking buffer overnight at 4 °C. Atg7 (AbCam Ab133528, 1:1000), Egln1 (Cell-Signaling Technology D31E11, 1:1000), p62/Sqstm1 (AbCam Ab109012, 1:1000), and Alpha-tubulin (Sigma-Aldrich Clone DM1, 1:2000) primary antibodies were used. Anti-mouse or anti-rabbit IRDye (LI-COR Biosciences) secondary antibodies and the Odyssey CLx Imager (LI-COR Biosciences) were utilized for detection and imaging.

## Fluorescent LC3 reporter assay

The FUW mCherry-GFP-LC3 reporter plasmid was obtained from Addgene (Plasmid #110060) which was utilized to generate lentivirus in 293 T cells. B16-F10 Cas9-expressing cells were then infected and grown for 48 h. Cells were then plated and treated with standard DMEM with 10% FBS or Hanks Balanced Salt Solution for the indicated time. Following the time course all cells were washed 3x with PBS and then resuspended in 100 uL of PBS and stained with LIVE/DEAD Fixable Violet (ThermoFisher L34955) stain per the manufacturer's instructions.

Cells were then passed through a 40-micron filter to generate a single cell suspension and then run on a BD FACSCelesta. Single cells were gated for live cells and then the individual GFP and mCherry signal were measured in the FITC and CF594 channels, respectively. Data were analyzed in FlowJo v10.

## Statistics and reproducibility

Statistical analyses were performed in the R with Bioconductor, and tables and plots were generated using dplyr[58] and ggplot2[59] packages. All cancer type abbreviations follow TCGA standards (https://gdc.cancer.gov/resources-tcga-users/tcga-code-tables/tcga-study-abbreviations). All box plots follow the same formatting, where the middle line indicates the median, the hinges represent the 25th and 75th percentile, notches indicate the 95% confidence interval, and whiskers represent the most extreme data point within 1.5x the interquartile range from hinge Sample sizes for in vivo validation experiments were based off power analysis to reliably detect a 25% reduction in tumor growth with greater than 90% confidence incorporating measured standard error rates from previous in vivo experiments[30].

## Reporting summary

Further information on research design is available in the Nature Portfolio Reporting Summary linked to this article.

## Data availability

Our proximal-Poly(A) KO (pKO) library has been deposited with Addgene (Pooled Library #81543 [https://www.addgene.org/Robert_Bradley/]). RNA-seq data generated as part of this study has been deposited in the Gene Expression Omnibus (accession number GSE212278). Gene-level 3′ UTR measurements for all TCGA samples were downloaded from http://tc3a.org[51]. Source data are provided with this paper. Data for Figs. 1–2, 4–5 and Supplementary Figs. 1–3, 5, 7–8 are included in Supplementary Data 1–5 and the Source Data file. All remaining data can be found in the Article, Supplementary and Source Data files. Source data are provided with this paper.

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

## Acknowledgements

We thank A. Rajan for feedback and advice on autophagy-related studies. We thank D. Bennett and the Wellcome Trust Functional Genomics Cell Bank for providing the Melan-A cells. A.M.G is an ARCS Foundation scholar. R.K.B. was supported in part by the NIH/NCI (R01 CA251138), NIH/NHLBI (R01 HL128239, R01 HL151651) and the Blood Cancer

Discoveries Grant program through the Leukemia & Lymphoma Society, Mark Foundation for Cancer Research, and Paul G. Allen Frontiers Group (8023-20). R.K.B is a Scholar of The Leukemia & Lymphoma Society (1344-18) and holds the McIlwain Family Endowed Chair in Data Science. Computational studies were supported in part by FHCC's Scientific Computing Infrastructure (ORIP S10 OD028685). Experimental studies were supported in part by the Experimental Histopathology, Flow Cytometry, and Genomics Shared Resources of the Fred Hutch/University of Washington Cancer Consortium (NIH/NCI P30 CA015704). The results in this publication are based in part on data from The Cancer Genome Atlas Research Network (http://cancergenome.nih.gov).

## Author contributions

AMG, JDT and RKB designed the study. AMG, JDT, AEB, SAM, TRN, TB and EC conducted experiments. AMG analyzed the experimental data and performed genomic analyses. AMG and RKB wrote the manuscript, with input from all authors.

## Competing interests

R.K.B. is a founder and scientific advisor of Codify Therapeutics and Synthesize Bio and holds equity in both companies. R.K.B. has received research funding from Codify Therapeutics unrelated to the current work. The remaining authors declare no competing interests.
