## [Peer Review File · Nature Communications]

Reviewers' Comments:

Reviewer #1:

Remarks to the Author:

In this manuscript, Gabel and colleagues quantify alternative polyadenylation (APA) across a range of melanoma RNAseq samples derived from TCGA. They find that, generally, longer 3' UTRs, corresponding to distal APA events, are associated with poor patient outcomes. They then design and implement a CRISPR screen aimed at identifying specific APA events that drive these outcomes. Using an in vitro cellular system and a mouse tumor graft assay, they identify several such events and follow up on APA in one gene: Atg7. CRISPR-mediated ablation of the proximal APA site in Atg7 forces usage of the distal APA site. This was associated with decreased cell viability in vitro and a corresponding increase in survival of mice engrafted with edited tumorigenic cells, demonstrating specific phenotypes derived from a specific APA event. Satisfyingly, this was followed up with an analysis of Atg7 APA isoform abundance as a function of patient survival in TCGA data where it was found that patients with predominant long Atg7 UTR isoforms had better outcomes.

Overall, this is a well-written manuscript that significantly advances the field. Many studies have been done in which APA was quantified in tumor samples and correlations were made with patient survival or other phenotypes. However, very few have been able to tie these phenotypes to APA outcomes in specific genes. When combined with the creation of a generalized high-throughput method for identifying such phenotype/isoform connections, this manuscript represents an important advance. I have only a few suggestions below that may improve the clarity of the manuscript and/or the strength of the conclusions.

MAJOR COMMENTS

1. Conclusions about the function and relevance of Atg7 APA isoforms could potentially be strengthened through the use of rescuing transgenes. As it is now, conclusions are drawn either from the relative abundance of cells in the screen or from later experiments comparing pgRNA-treated cells to control cells.
2. The data presented in figure 3D seems somewhat weak. Although I trust the authors performed quantification of intensities within the gel, it is hard for me to conclude by eye that distal APA is increased in the pKO samples. Perhaps this is due to the nested RT-PCR strategy. The use of a Northern blot may be better here as its lack of amplification may reduce off-target noise bands while still having potentially increased sensitivity relative to RT-PCR (if, for example, radioactive probes are used). Regardless, another way to quantify APA in these samples would be preferred.

MINOR COMMENTS

1. The x-axis label of figure 1B could be more informative. It is explained somewhat in the legend to figure 1A. APA experts may pick up on what it is, but people outside this specialized subfield may not.
2. In figure 3E, the y-axis is labeled as a percent, but is very likely not a percent but rather a fraction. If labeled as a percent, it should likely go from 10% to 50% as presented.
3. Figures 1D through 1H are not referenced in the text (unless I missed them).
4. In describing figure 2C, the authors state that "B16-F10 cells more preferentially expressed SAP30L mRNA with the shorter 3' UTR, while Melan-A cells preferentially utilized the longer 3' UTR". While Melan-A cells do use the distal APA site more than the B16-F10 cells, they still do prefer the proximal APA site (see also the $\log_2(\text{distal/proximal})$ value being < 0 in figure 3E). This statement should therefore be rephrased. Perhaps it could be changed to state that "B16-F10 cells more preferentially expressed SAP30L mRNA with the shorter 3' UTR, while Melan-A cells more preferentially utilized the longer 3' UTR."

Reviewer #2:

Remarks to the Author:

In this manuscript, Gebel et al. study the link between alternative polyadenylation in 3' untranslated regions (3' UTR) and tumor growth and/or aggressiveness.

They start with describing the alternative polyadenylation (i.e. the length of the 3'UTR) landscape in tumors, by analyzing data from the TCGA with already reported bioinformatic methods (Fig. 1). Depending on the tumor type, they observed an association between worse overall survival and either a shortening (e.g. ovarian carcinoma) or a lengthening (e.g. melanoma) of the 3'UTR depending on the tumor types.

They went on to describe APA in murine immortalized melanocytes and highly aggressive melanoma cells (B16-F10) (Fig. 2). They found widespread shortening or lengthening of the 3'UTR. They then deleted proximal polyadenylation site (PAS) for 143 candidate genes in the B16-F10 cell line (Fig. 3 and 4) and performed a screen to detect events associated with growth of B16-F10 cells in vitro and in mice. They next focused on Atg7 (Fig. 4 and 5) for which KO of the proximal PAS reduced cell/tumor growth but had no effect on autophagy (as previously described).

Major points

-The authors always conclude on APA regulation but, in most of their analysis, the difference in 3'UTR length could also be due to differential mRNA stability. This has to be experimentally addressed in Fig. 3 and Fig. 5. This point also has to be discussed to tone down the conclusions they raised on tumor samples.

-The authors analyzed overall survival in patients. This is for most cancer types (and more specifically in melanoma) linked to tumor dissemination, and not tumor growth. All the in vitro and in vivo experiments must therefore be repeated by using cell migration/invasion (and formation of metastasis in mice) as a read out, instead of just cell/tumor growth. The B16-F10 cells that are highly invasive and able to form metastasis in mice are actually designed for such analysis.

-Authors must perform a global analysis of the length of the 3'UTR in the different B16-derived cell lines with different invasive properties. This will be more informative and more related to the analysis they performed with human tumors than comparing B16-F10 with immortalized melanocytes.

-The function of the genes with differential 3'UTR in tumors and in cell lines should be examined and commented.

-The title doesn't reflect the conclusions raised in this ms. They do not link the process of alternative polyadenylation, but the length of 3'UTR, to a phenotype. Only tumor growth is analyzed and not "tumor phenotypes". This is actually what is missing in this ms. While the link between 3'UTR length and cell growth is well established, the connection between 3'UTR length and tumor cell spreading remains to be firmly established.

Reviewer #3:

Remarks to the Author:

Almost a third of genes have >1 polyA site and therefore can generate mRNA transcripts with longer or shorter 3'UTR because of alternate polyadenylation.

This is an interesting study of how these longer or shorter 3'UTR can increase or decrease survival in specific cancer types. To understand the mechanisms behind this observation, the authors used two cellular systems – the syngeneic B16-F10 melanoma mouse model (for in vivo studies) and the Melan-A melanocyte cell line derived from the C57BL/6 mouse (for in vitro studies).

They generated a CRISPR library targeting for deletion proximal polyA sites in approx. 300 mouse

genes (including as controls 15 specific genes known to be involved in tumour growth) in order to force B16-F10 or Melan-A cells following transduction to use distal polyA sites and generate transcripts with longer 3'UTR for that specific gene. This may affect mRNA stability and expression levels.

At day 20 post transduction, sequencing was used to compare pgRNA levels with that at day 0, to detect enrichment (increased cell growth) vs depletion (impaired viability or growth). The gene ATG7 was selected as depleted at day 20 in both the in vitro and in vivo experiments. Surprisingly, while complete KO of ATG7 caused increased p62 levels (a classical reporter of autophagy), forced distal PA usage did not increase autophagy and only reduced expression by 60%, although did increase the fraction of cells in S and G2/M phases of the cell cycle.

General comments

For all the Supp Tables, it would be useful to insert 4-5 lines above each table with a description of what the different column headers mean e.g. in Supp Table 5, it takes a bit of guesswork to get to what "fc_norm|B16|startall|vitroall|1A|1A" actually means. This would save the reader time, which I know is precious for all of us.

Specific comments

1. Introduction section,

Many of the readers will be less familiar with the mechanics of polyadenylation of mRNA and the introduction would benefit from a clearer description of how genes can contain >1 polyA site leading to different cleavage and variable lengths of 3'-UTR.

2. In Results section, "For each cancer subtype, we stratified patients into terciles representing whether their tumor transcriptomes preferentially expressed short, medium, or long 3' UTRs by computing a median 3' UTR length for each tumor across 7,513 genes that are subject to alternative polyadenylation (Figure 1A-H; Supp. Figure 1A-C)."

Can the authors comment on why this approach was used and what others were considered as a way to define a 'median' UTR?

3. In Results section, "The final library comprises 8-10 unique targeting pgRNAs per proximal poly(A) signal, 150 positive control pgRNAs designed to knock out 15 distinct genes known to be involved in tumor growth control, and 150 negative control pgRNAs targeting proximal polyadenylation signals..."

Which genes were selected as the 15 known to mediate tumour growth? These should be highlighted in as a Supp Table. Useful to compare how these performed against the depleted and enriched APA targets in 4C/D – the depleted in particular have a much greater negative LFC compared to the APA targets – why do you think this is the case?

The unexpressed controls have a very wide range of positive and negative values, more than I would have expected for controls. Why is this?

4. For the syngeneic CRISPR screen using the B16-F10 library transduced cells, were any analyses carried out to compare coverage of the library at an early timepoint versus day 0 (or plasmid)? CRISPR library representation can be heavily skewed by a small number of clones in cell line xenograft experiments. Usually a barcode expt would be carried out first to assess what complexity of library is likely to be maintained after transduced cells are engrafted.

5. For statistical analysis of custom pgRNA CRISPR library, "In brief, we normalized the fold-changes relative to day 0 for a given replicate such that the median of all pgRNAs targeting poly(A) sites in unexpressed genes was equal to 1 and pooled computed fold-changes across all replicates for a given condition to maximize statistical power"

Why was this approach adopted rather than simply comparing gRNA counts at Day 20 vs Day 0 for all pgRNA? Especially as the pgRNA targeting non-expressed genes in Fig 4C/D appear to have a significant viability effect.

6. ATG7 follow-up studies

Forced distal PA usage in ATG7 did reduce growth in vivo but did not increase p62 levels - were the authors able to show by PolyA-seq the expected change in length of 3'UTR following pKO with pgRNA, to help rule out off target effects of these pgRNA as an explanation for the observed phenotypes? Autophagy was ruled out as a mechanism of action of the ATG7 3'UTR lengthening on tumour cell growth in vivo using p62 expression and a LC3 reporter - are these sufficient to confidently exclude an autophagy effect here as a reason for the observed phenotype? Otherwise, we're left with changes in cell cycle, which is not a particularly satisfying mechanism of action.

REVIEWER COMMENTS

Reviewer #1 - APA, functional genomics (Remarks to the Author):

In this manuscript, Gabel and colleagues quantify alternative polyadenylation (APA) across a range of melanoma RNAseq samples derived from TCGA. They find that, generally, longer 3' UTRs, corresponding to distal APA events, are associated with poor patient outcomes. They then design and implement a CRISPR screen aimed at identifying specific APA events that drive these outcomes. Using an in vitro cellular system and a mouse tumor graft assay, they identify several such events and follow up on APA in one gene: *Atg7*. CRISPR-mediated ablation of the proximal APA site in *Atg7* forces usage of the distal APA site. This was associated with decreased cell viability in vitro and a corresponding increase in survival of mice engrafted with edited tumorigenic cells, demonstrating specific phenotypes derived from a specific APA event. Satisfyingly, this was followed up with an analysis of *Atg7* APA isoform abundance as a function of patient survival in TCGA data where it was found that patients with predominant long *Atg7* UTR isoforms had better outcomes.

Overall, this is a well-written manuscript that significantly advances the field. Many studies have been done in which APA was quantified in tumor samples and correlations were made with patient survival or other phenotypes. However, very few have been able to tie these phenotypes to APA outcomes in specific genes. When combined with the creation of a generalized high-throughput method for identifying such phenotype/isoform connections, this manuscript represents an important advance. I have only a few suggestions below that may improve the clarity of the manuscript and/or the strength of the conclusions.

REPLY: We thank this reviewer for their time, detailed summary of our work, and the kind words that “this manuscript represents an important advance”. We thank them for their thoughtful comments and questions, which we feel have notably improved the quality of our manuscript.

MAJOR COMMENTS

1. Conclusions about the function and relevance of *Atg7* APA isoforms could potentially be strengthened through the use of rescuing transgenes. As it is now, conclusions are drawn either from the relative abundance of cells in the screen or from later experiments comparing pgRNA-treated cells to control cells.

REPLY: We thank the reviewer for this comment. We agree that a rescue experiment is an excellent way to test hypotheses arising from functional experiments. We therefore followed the reviewer’s suggestion and include new data demonstrating that restoration of *ATG7* protein rescues the reduced growth phenotype that we observed in *Atg7* pKO cells (in which we ablated the gene-proximal poly(A) site):

- As the *Atg7* pKO cell line is a specific knock-out of the short *Atg7* isoform, we used this cell line as a background for our rescue experiments. We generated two *Atg7* cDNA constructs that harbored identical *Atg7* coding sequences and either an SV40 poly(A) signal directly downstream of the stop codon or the endogenous *Atg7* proximal 3' UTR including 200 nucleotides after the proximal poly(A) signal sequence (see schematic below and **Supp. Figure 8A**). We then validated that stable integration of the cDNA constructs rescued the reduced protein level present in *Atg7* pKO cells (**Supp. Figure 8B**). We then measured cell growth *in vitro* and found that introduction of either the *Atg7* SV40 or *Atg7* short 3' UTR cDNA rescued the reduced cellular proliferation phenotype present in *Atg7* pKO cells (**Supp. Figure 8C**).
- We observed complete rescue with the SV40 construct and partial rescue with the short 3' UTR construct. As the SV40 construct led to higher ATG7 protein levels, these data are consistent with a model where reduced ATG7 protein levels, rather than loss of the short 3' UTR isoform per se, in *Atg7* pKO cells are the primary driver of the reduced cell growth phenotype of those cells (with the caveat, which is unavoidable in such rescue experiments, that transgenic expression of a gene in a rescue experiment is not fully equivalent to expression from the endogenous locus).

Transgenic restoration of ATG7 protein rescues the *Atg7* pKO growth phenotype.

(A) Schematic of two *Atg7* cDNA constructs harboring identical coding sequences, but distinct 3' UTRs, with either an SV40 poly(A) signal or the endogenous 3' UTR and the proximal poly(A) site.

(B) Immunoblot of protein collected from Cas9-expressing B16-F10 cells treated with either a control pgRNA or *Atg7* pKO pgRNA with stable expression of the indicated *Atg7* cDNA construct. Protein ratio normalized to α -Tubulin concentration and then to the control pKO protein ratio.

(C) *In vitro* cell growth of Cas9-expressing B16-F10 cells treated with a control pgRNA or *Atg7* pKO pgRNA with the indicated cDNA constructs as measured by CellTiter-Glo. Measurement is the average of three replicates \pm standard error of the mean.

2. The data presented in figure 3D seems somewhat weak. Although I trust the authors performed quantification of intensities within the gel, it is hard for me to conclude by eye that distal APA is increased in the pKO samples. Perhaps this is due to the nested RT-PCR strategy. The use of a Northern blot may be better here as its lack of amplification may reduce off-target noise bands while still having potentially increased sensitivity relative to RT-PCR (if, for example, radioactive probes are used). Regardless, another way to quantify APA in these samples would be preferred.

REPLY: We thank the reviewer for this comment and understand that the noise of the gel makes its interpretation less obvious than ideal. To remedy this, we completed Poly(A) sequencing of all polyclonal cell lines generated for this manuscript. As compared to RT-PCR or other targeted measures, Poly(A)-seq allows for transcriptome-wide quantification of thousands of APA events. This allows us to simultaneously quantify the on-target efficacy of a specific pgRNA, identify use of non-canonical poly(A) sites within the target transcript, and screen genome-wide for off-target APA events that occur following pgRNA treatment. The Poly(A)-seq data for both *Sap30l* pKO pgRNAs show striking and significant shifts towards use of the distal poly(A) site relative to the control sample, indicating that pgRNA treatment leads to enhanced use of distal poly(A) sites as desired (revised **Figure 3D**). Looking genome-wide, we clearly identify *Sap30l* as the most significantly altered APA event in *Sap30l* pKO-treated cells, confirming specificity of this pgRNA for altering APA (**Supp. Figure 4C**).

We additionally performed poly(A)-seq for the control, *Egln1* pKO, and *Atg7* pKO polyclonal cell lines. These experiments revealed specific, increased use of distal poly(A) sites reflecting gene-specific 3' UTR lengthening that is highly concordant with the RT-PCR data presented in the previous version of this manuscript (**Figure 4E** and **4H**). Seen below are Poly(A)-seq data for each cell line indicated:

Poly(A)-seq of all polyclonal pKO cell lines used.

BAM coverage plots of Poly(A)-seq data generated per the indicated B16-F10 Cas9 cell line treated with control, Sap30l_1, Sap30l_2, Atg7 or EglN1 proximal poly(A) KO (pKO) pgRNAs display decreased proximal poly(A) site use and increased distal poly(A) site use, as expected.

MINOR COMMENTS

1. The x-axis label of figure 1B could be more informative. It is explained somewhat in the legend to figure 1A. APA experts may pick up on what it is, but people outside this specialized subfield may not.

REPLY: We thank the reviewer for this note and have added additional detail accordingly.

2. In figure 3E, the y-axis is labeled as a percent, but is very likely not a percent but rather a fraction. If labeled as a percent, it should likely go from 10% to 50% as presented.

REPLY: We thank the reviewer for this note and have changed it accordingly.

3. Figures 1D through 1H are not referenced in the text (unless I missed them).

REPLY: We thank the reviewer for this note. We added appropriate figure references.

4. In describing figure 2C, the authors state that “B16-F10 cells more preferentially expressed SAP30L mRNA with the shorter 3’ UTR, while Melan-A cells preferentially utilized the longer 3’ UTR”. While Melan-A cells do use the distal APA site more than the B16-F10 cells, they still do prefer the proximal APA site (see also the $\log_2(\text{distal/proximal})$ value being < 0 in figure 3E). This statement should therefore be rephrased. Perhaps it could be changed to state that “B16-F10 cells more preferentially expressed SAP30L mRNA with the shorter 3’ UTR, while Melan-A cells more preferentially utilized the longer 3’ UTR.”

REPLY: We thank the reviewer for this note and have changed it accordingly.

Reviewer #2 - RNA processing, melanoma (Remarks to the Author):

In this manuscript, Gebel et al. study the link between alternative polyadenylation in 3' untranslated regions (3' UTR) and tumor growth and/or aggressiveness.

They start with describing the alternative polyadenylation (i.e. the length of the 3'UTR) landscape in tumors, by analyzing data from the TCGA with already reported bioinformatic methods (Fig. 1). Depending on the tumor type, they observed an association between worse overall survival and either a shortening (e.g. ovarian carcinoma) or a lengthening (e.g. melanoma) of the 3'UTR depending on the tumor types.

They went on to describe APA in murine immortalized melanocytes and highly aggressive melanoma cells (B16-F10) (Fig. 2). They found widespread shortening or lengthening of the 3'UTR. They then deleted proximal polyadenylation site (PAS) for 143 candidate genes in the B16-F10 cell line (Fig. 3 and 4) and performed a screen to detect events associated with growth of B16-F10 cells in vitro and in mice. They next focused on *Atg7* (Fig. 4 and 5) for which KO of the proximal PAS reduced cell/tumor growth but had no effect on autophagy (as previously described).

REPLY: We thank the reviewer for their time and comments.

Major points

-The authors always conclude on APA regulation but, in most of their analysis, the difference in 3'UTR length could also be due to differential mRNA stability. This has to be experimentally addressed in Fig. 3 and Fig. 5. This point also has to be discussed to tone down the conclusions they raised on tumor samples.

REPLY: We thank the reviewer for this comment and important suggestion that mRNA stability may be altered by APA. We have completed additional experiments to address this suggestion by measuring stability of the short and long *Atg7* isoforms.

- Consistent with prior work in the field of 3' UTR regulation, the data presented in this manuscript indicates that increased distal poly(A) site use for *Atg7* leads to reduced ATG7 mRNA abundance and lower ATG7 protein levels (**Figure 4F**). This is in line with several publications demonstrating that 3' UTR lengthening that occurs as a result of distal poly(A) site use can destabilize mRNAs via multiple mechanisms, including increased abundance of potential regulatory binding sites for miRNAs or RNA-binding proteins (Mayr & Bartel 2009, *Cell*; Jenal et al 2012, *Cell*; Mufteev et al 2023, *Biorxiv*). There is also a weak, but significant, association between distal poly(A) site use and mRNA abundance (Jenal et al 2012, *Cell*; Goering et al 2021, *BMC Genomics*), which many have speculated is a general trend whereby distal poly(A) site use, and therefore 3' UTR lengthening, tends to destabilize mRNA. However, it is also important to

recognize these global correlations remain under active study and such trends may or may not hold for individual genes (Gruber et al 2014, *Nature Communications*).

- To experimentally test the association between *Atg7* poly(A) site use and *Atg7* mRNA stability, we performed isoform-specific RT-PCR to measure *Atg7* mRNA levels B16-F10 cells over an Actinomycin D treatment time course. These experiments permitted us to quantitatively assess the relative mRNA stabilities of the short and long isoforms of *Atg7* using the percent distal isoform usage metric. These data revealed that the *Atg7* percent distal isoform decreases rapidly over a 24h time course, consistent with our hypothesis that the *Atg7* isoform utilizing the distal poly(A) site is less stable than is the isoform utilizing the proximal poly(A) site (**Figure 5A-B**).

-The authors analyzed overall survival in patients. This is for most cancer types (and more specifically in melanoma) linked to tumor dissemination, and not tumor growth.

REPLY: We thank the reviewer for pointing out that tumor growth and tumor dissemination are interesting and distinct phenotypes that each warrant study. We agree that metastasis is often linked to overall survival. In the revised manuscript, we include analyses of both progression-free survival and overall survival, which reveal that many of the associations between APA and patient survival that motivated our original study persist for progression-free survival as well, which is often associated with growth of the primary tumors (**Supp. Figure 2C**). We also note the important point that overall survival is frequently linked to metastasis in new text in the Discussion (see below).

All the in vitro and in vivo experiments must therefore be repeated by using cell migration/invasion (and formation of metastasis in mice) as a read out, instead of just cell/tumor growth. The B16-F10 cells that are highly invasive and able to form metastasis in mice are actually designed for such analysis.

REPLY: As increased relative expression of the *Atg7* long 3' UTR isoform is associated with prolonged overall survival in human melanoma, we reasoned the B16-F10 *Atg7* pKO cell line might provide a useful system to begin to assess how APA modulation influenced cell migratory potential. We performed scratch assays to compare the migration rate of control pKO and *Atg7* pKO cells. These experiments revealed that *Atg7* pKO cells migrated into the space significantly slower than did our negative control (*Crabp1*) pKO cells, suggesting that loss of the *Atg7* short 3' UTR isoform reduced migratory potential.

We agree with the reviewer that further studies focused on cell migration and metastasis would be valuable experiments. However, given that the current manuscript is focused on creating a methodological approach for high-throughput characterization of APA in the context of tumor growth and demonstrating the feasibility of using this method to identify specific APA events of functional relevance, we respectfully believe that additional investigation of metastatic phenotypes is beyond the scope of this manuscript. We hope that future studies apply our or other approaches to dissect the role of APA in tumor dissemination and have added the following text to the Discussion to highlight the scope of the current study and the need for future work focused on metastasis:

It is important to note that our *in vivo* studies focused on determining of how APA modulation altered tumor cell growth, which is only one of many important cancer phenotypes that influence patient prognosis. For example, overall survival is frequently strongly associated with metastasis, which our assays did not measure. A natural and important extension of the current study would be to determine how cancer-specific APA influenced tumor cell dissemination.

-Authors must perform a global analysis of the length of the 3'UTR in the different B16-derived cell lines with different invasive properties. This will be more informative and more related to the analysis they performed with human tumors than comparing B16-F10 with immortalized melanocytes.

REPLY: While such an experiment would be interesting, we believe that characterizing the landscape of other B16-derived cell lines based on their invasive properties is beyond the scope of our current study, which is focused on tumor growth rather than metastasis. The major innovations in our work are implementing a high-throughput, CRISPR/Cas9 paired-guide RNA screening platform to functionally study poly(A) site selection and demonstrating the potential of this platform to identify specific APA events of functional relevance. We hope that this method is adopted by others to study additional biological questions in the future, including how APA influences metastasis.

-The function of the genes with differential 3'UTR in tumors and in cell lines should be examined and commented.

REPLY: We thank the reviewer for this great suggestion. We conducted such a gene function analysis by focusing on genes that exhibited highly correlated 3' UTR lengths between human clinical melanoma and mouse melanoma, as the cross-species nature of such genes suggests potential functional relevance. We performed a Gene Ontology (GO) enrichment analysis on this gene set, which revealed many enriched GO terms, notably including many terms related to RNA processing itself (**Supp. Figure 3C**).

-The title doesn't reflect the conclusions raised in this ms. They do not link the process of alternative polyadenylation, but the length of 3'UTR, to a phenotype. Only tumor growth is analyzed and not "tumor phenotypes". This is actually what is missing in this ms. While the link between 3'UTR length and cell growth is well established, the connection between 3'UTR length and tumor cell spreading remains to be firmly established.

REPLY: We thank the reviewer for this comment. We revised the title to "Highly multiplexed screening reveals how cancer-specific alternative polyadenylation shapes tumor growth *in vivo*" in order to accurately reflect the focus of the manuscript on tumor growth. We respectfully submit that the phrase "alternative polyadenylation" is accurate, as a shift in patterns of alternative polyadenylation necessarily alters the patterns of expressed 3' UTRs.

Reviewer #3 - CRISPR Screens (Remarks to the Author):

Almost a third of genes have >1 polyA site and therefore can generate mRNA transcripts with longer or shorter 3'UTR because of alternate polyadenylation.

This is an interesting study of how these longer or shorter 3'UTR can increase or decrease survival in specific cancer types. To understand the mechanisms behind this observation, the authors used two cellular systems – the syngeneic B16-F10 melanoma mouse model (for in vivo studies) and the Melan-A melanocyte cell line derived from the C57BL/6 mouse (for in vitro studies).

They generated a CRISPR library targeting for deletion proximal polyA sites in approx. 300 mouse genes (including as controls 15 specific genes known to be involved in tumour growth) in order to force B16-F10 or Melan-A cells following transduction to use distal polyA sites and generate transcripts with longer 3'UTR for that specific gene. This may affect mRNA stability and expression levels.

At day 20 post transduction, sequencing was used to compare pgRNA levels with that at day 0, to detect enrichment (increased cell growth) vs depletion (impaired viability or growth). The gene ATG7 was selected as depleted at day 20 in both the in vitro and in vivo experiments. Surprisingly, while complete KO of ATG7 caused increased p62 levels (a classical reporter of autophagy), forced distal PA usage did not increase autophagy and only reduced expression by 60%, although did increase the fraction of cells in S and G2/M phases of the cell cycle.

REPLY: We thank the reviewer for the time and energy invested in their review and in particular their insightful comments and questions regarding the mechanistic underpinnings and methodology of our CRISPR screening approach, which helped us to notably improve our manuscript.

General comments

For all the Supp Tables, it would be useful to insert 4-5 lines above each table with a description of what the different column headers mean e.g. in Supp Table 5, it takes a bit of guesswork to get to what “fc_norm|B16|startall|vitroall|1A|1A” actually means. This would save the reader time, which I know is precious for all of us.

REPLY: We thank the reviewer for this important note and have added better descriptions within the tables.

Specific comments

1. Introduction section,

Many of the readers will be less familiar with the mechanics of polyadenylation of mRNA and the introduction would benefit from a clearer description of how genes can contain >1 polyA site leading to different cleavage and variable lengths of 3'-UTR.

REPLY: We thank the reviewer for this helpful suggestion. We revised the introduction to add additional background, and the text now reads:

Alternative cleavage and polyadenylation (APA) – the process whereby a pre-mRNA can be cleaved and have a poly(A) tail added at multiple distinct locations, leading to expression of mRNAs with different 3' untranslated regions (UTRs) – affects most human genes (1). Although the biological roles of most APA events are unknown, detailed studies have revealed that APA can affect mRNA levels, localization, and translation, among other molecular phenotypes (2-4). Poly(A) site selection is coordinated by differential interactions between several RNA-binding proteins and the U1 small nuclear RNA (snRNA). RNA-binding proteins and the U1 snRNA bind RNAs in a sequence specific manner near the proximal poly(A) site and can either promote or inhibit assembly of the 3' end processing complex, though this process is still not entirely understood (5-8). Differential polyadenylation site usage is commonly observed in many biological processes, frequently in a biased manner indicative of coordinated regulatory changes (9-11). For example, rapidly dividing cells frequently utilize gene-proximal over gene-distal poly(A) sites, and thus express mRNAs with correspondingly shorter 3' UTRs, compared to non-dividing, terminally differentiated cells. mRNAs are differentially polyadenylated throughout development, where terminally differentiated cells tend to utilize more distal poly(A) sites (3); immune cell subsets including monocytes, T cells, and B cells undergo global 3' UTR shortening when stimulated by their respective chemokines to begin dividing (12); many cancers express mRNAs with markedly shorter 3' UTRs than do peritumoral, healthy tissues (13-15).

2. In Results section, “For each cancer subtype, we stratified patients into terciles representing whether their tumor transcriptomes preferentially expressed short, medium, or long 3' UTRs by computing a median 3' UTR length for each tumor across 7,513 genes that are subject to alternative polyadenylation (**Figure 1A-H; Supp. Figure 1A-C**).”

Can the authors comment on why this approach was used and what others were considered as a way to define a 'median' UTR?

REPLY: The major advantage of our approach is that it allowed us to analyze datasets that lack sufficient numbers of samples from healthy tissues to stratify patients as preferentially using short versus long 3' UTRs by comparing tumor samples with healthy tissues. Many landmark studies in this field have relied upon comparing healthy and tumor samples in this manner, but this has necessarily limited those studies to analyzing the subset of tumor types for which sufficient healthy samples are available. Specifically, the TCGA contains only 12 (of 31 total) cancer cohorts with more than 15 normal matched tissues and only 17 with any matched

normals at all. There are other potential approaches to patient stratification that we could have undertaken, such as normalizing each tumor samples per cancer subtype to the median 3' UTR length per gene across all normal samples (Xia. et al 2012; Goering et al 2021). However, these approaches also would not allow analysis of all cancer subtypes, including melanoma samples, where adjacent healthy tissue samples were not available.

We have edited our text to clarify this point, and the revised text now reads:

Several prior studies have analyzed RNA sequencing data from The Cancer Genome Atlas (TCGA) and observed significant associations between global APA dysregulation and patient survival; however, those studies limited analyses to the 17 tumor types for which there were sufficient patient-matched, peritumoral normal tissue samples available (9, 11). In order to extend such analyses to all tumor types, including those for which peritumoral samples are not available, we instead took a stratification-based approach that relied on data from tumor samples alone. For each cancer subtype, we stratified patients into terciles representing whether their tumor transcriptomes preferentially expressed short, medium, or long 3' UTRs by computing a median 3' UTR length for each tumor across 7,513 genes that are subject to alternative polyadenylation (**Figure 1A-H; Supp. Figure 1A-C**).

3. In Results section, “The final library comprises 8-10 unique targeting pgRNAs per proximal poly(A) signal, 150 positive control pgRNAs designed to knock out 15 distinct genes known to be involved in tumor growth control, and 150 negative control pgRNAs targeting proximal polyadenylation signals...”

Which genes were selected as the 15 known to mediate tumour growth? These should be highlighted in as a Supp Table.

REPLY: We thank the reviewer for this comment and have edited the text accordingly.

- The previous wording in our manuscript “Genes known to modify tumor growth” is a bit ambiguous, and we have reworded this to be more specific. These are either genes known to kill most cells via DepMap (random sample of core essential gene list) or have been identified as promoting enhanced cell growth *in vitro* or *in vivo* by mining B16-F10 CRISPR-Cas9 genome wide gene knockout screen data reported by another publication (Manguso et al 2020, *Nature*). For example, by analyzing supplementary data files from that manuscript, we identified that the gene *Nf2* enhanced B16-F10 growth significantly *in vivo* relative to *in vitro*. (See Manguso et al 2020, Supplementary Table 1, Sheet Titled TCRaKO v *in vitro* enriched: *Nf2* is the top enriched hit, suggesting growth *in vivo* (though a T cell deficient system) is greatly enhanced relative to growth of cells *in vitro*, as we also observed in our model system).
- Per the above point, we clarify the selection of the 15 genes with the following new text in the revised manuscript: “An additional 15 genes were included as growth controls based on previous literature, either core essential genes identified from DepMap or genes which promote more rapid B16-F10 cell growth *in vivo* identified by mining data

generated from previous genome wide gene knock out CRISPR screens in B16-F10 cells (60).”

- We have also highlighted these genes within Supplementary table 4 via the “target_source” column which is now more clearly defined in the legend contained in the supplementary table.

Useful to compare how these performed against the depleted and enriched APA targets in 4C/D – the depleted in particular have a much greater negative LFC compared to the APA targets – why do you think this is the case?

These positive controls were purposefully designed as gene knockouts targeting core essential genes. As such, they represent the most extreme effects on fitness that we can observe in our screen and are expected to have deleterious effects on cell viability that are generally stronger than what we expect from the more modest genomic perturbation of deleting single alternative polyadenylation sites.

The unexpressed controls have a very wide range of positive and negative values, more than I would have expected for controls. Why is this?

REPLY: We thank the reviewer for this important question. We have included a detailed response below with additional analyses.

One important consideration that likely contributes to the increased variance that we observed for our control pgRNAs targeting unexpressed genes is that gRNA selection is much more constrained when targeting poly(A) sites than when simply disrupting a coding sequence or deleting larger genomic elements. This constraint means that we (and others manipulating endogenous poly(A) sites) must necessarily utilize occasionally suboptimal gRNAs. 3' UTRs are significantly more AT rich than are coding regions (Louie et al 2003, *Genome Research*), leading to reductions in the availability of high quality gRNAs (high on-target efficiency, low off-target efficiency) as optimal GC content is an important predictor of gRNA efficiency (Wang et al 2014, *Science*; Gagnon et al 2014, *PLoS ONE*; Doench et al 2014, *Nature Biotechnology*). This difficulty is compounded by the fact that our poly(A) site targeting libraries are spatially restricted to a 200 bp window around the poly(A) site. We artificially imposed this 200 bp window to minimize the potential effects of deleting larger genomic windows while still having the ability to generate >8 decent quality pgRNAs per target. In all, this constitutes a significantly smaller search space to select gRNAs from compared to more commonly used gene knockout libraries, which can select gRNAs from the entire coding sequence. Taken together, these factors mean any individual gRNA pulled from our library is likely to be less efficient and less specific as a function of what we are targeting. In order to experimentally account for a likely increased rate of potential false positive hits due to off-target effects of suboptimal gRNA use, we subsampled all possible generated control pgRNAs targeting proximal poly(A) sites in unexpressed genes in B16-F10 cells so that the distribution of on-target and off-target scores was similar to our pKO library targeting poly(A) sites in expressed

genes (Supp. Figure 5C-D). This is the likely explanation for a higher variance in effect sizes of our controls in our library (compared to what is typically seen in high-throughput CRISPR/Cas9

screens, where control gRNAs are often non-targeting and thus do not induce any DNA damage and / or have very low off-target scores). See below the density plots of the distribution of our gRNAs comprising our library compared to the efficiency (Rule Set 2 score) and specificity (Guidescan specificity score) of the optimized CRISPR/Cas9 Brie library, which is a genome-wide, gene-knockout library for mouse (Sanson et al 2018, *Nature Communications*).

We included an unusually large number of negative controls (~10% of the library compared to gene KO libraries like the Brie library, which are ~1-2% controls) in order to increase our statistical power in the face of the unavoidable constraints associated with manipulating poly(A) sites with CRISPR.

Given the challenge of selecting optimal gRNAs for manipulating endogenous poly(A) sites, we utilized genome-wide assays to confirm the specificity of targeting for key pgRNAs that we

studied in the manuscript. For example, we performed Poly(A) sequencing of all polyclonal cell lines generated for this manuscript in order to demonstrate the expected shifts in poly(A) site usage following pgRNA introduction. As compared to RT-PCR or other targeted measures, Poly(A)-seq allows for transcriptome-wide quantification of thousands of APA events. This allows us to simultaneously quantify the on-target efficacy of a specific pgRNA, identify use of non-canonical poly(A) sites within the target transcript, and screen genome-wide for off-target APA events that occur following pgRNA treatment. The Poly(A)-seq data for both Sap30l pKO pgRNAs show striking and significant shifts towards use of the distal poly(A) site relative to the control sample, indicating that pgRNA treatment leads to enhanced use of distal poly(A) sites as desired (revised **Figure 3D**). Looking genome-wide, we clearly identify Sap30l as the most significantly altered APA event in Sap30l pKO-treated cells, confirming specificity of this pgRNA for altering APA (**Supp. Figure 4C**).

We additionally performed poly(A)-seq for the control, *Egln1* pKO, and *Atg7* pKO polyclonal cell lines. These experiments revealed specific, increased use of distal poly(A) sites reflecting gene-specific 3' UTR lengthening that is highly concordant with the RT-PCR data presented in an earlier version of this manuscript (**Figure 4E** and **4H**). Seen below are Poly(A)-seq data for each cell line indicated:

Poly(A)-seq of all polyclonal pKO cell lines used.

BAM coverage plots of Poly(A)-seq data generated per the indicated B16-F10 Cas9 cell line treated with control, Sap30l_1, Sap30l_2, Atg7 or Egl1 proximal poly(A) KO (pKO) pgRNAs display decreased proximal poly(A) site use and increased distal poly(A) site use, as expected.

4. For the syngeneic CRISPR screen using the B16-F10 library transduced cells, were any analyses carried out to compare coverage of the library at an early timepoint versus day 0 (or plasmid)? CRISPR library representation can be heavily skewed by a small number of clones in

cell line xenograft experiments. Usually a barcode expt would be carried out first to assess what complexity of library is likely to be maintained after transduced cells are engrafted.

REPLY: We thank the reviewer for this important comment.

- Given the frequently high biological variability that characterizes *in vivo* tumorigenesis (as pointed out by the reviewer), we performed 8 replicates and pooled the sequencing data across replicates to minimize any potential issues with clone skewing in particular replicates or engraftment experiments. We performed all statistical analyses using this pooled data in order to maximize our statistical power. We additionally included an unusually large number of negative controls (~10% of the library compared to gene KO libraries like the Brie library, which are ~1-2% controls) in order to account for the inherently high variability associated with *in vivo* screening approaches.
- To address the reviewer's important question about library coverage, we directly analyzed pgRNA representation at the conclusion of the *in vivo* tumorigenesis experiment to assess library coverage and identify any potential issues with dropout. This analysis revealed that <1% of pgRNAs had read counts < 10, and the vast majority of all pgRNAs had counts > 1000, indicating that very little drop out occurred in our experiment.

- These above data are very similar to levels of drop out or significant over-enrichment observed in our previously published study that utilized a custom pgRNA library for *in vivo* screening in mice (Thomas et al 2021, *Nature Genetics*).
- In the below analyses, we broke out all pgRNAs based upon their classification as growth controls or pKO targeting pgRNAs. These analyses reveal that growth controls show significant drop out (as expected, since they are predicted to kill cells), while targeting pgRNAs show notably less drop out. The dotted line represents 10 counts; for each plot, the percentage of pgRNAs with < 10 counts is annotated.

5. For statistical analysis of custom pgRNA CRISPR library, “In brief, we normalized the fold-changes relative to day 0 for a given replicate such that the median of all pgRNAs targeting poly(A) sites in unexpressed genes was equal to 1 and pooled computed fold-changes across all replicates for a given condition to maximize statistical power”

Why was this approach adopted rather than simply comparing gRNA counts at Day 20 vs Day 0 for all pgRNA? Especially as the pgRNA targeting non-expressed genes in Fig 4C/D appear to have a significant viability effect.

REPLY: Thank you for this important question. This approach was adopted in our paper and in our previous publication (Thomas et al 2021, *Nature Genetics*) as normalizing the effect size to our control pgRNAs accounts for the deleterious effect of genomic DNA damage induced by gRNA cutting (Haapaniemi et al 2018, *Nature Medicine*). This procedure ensures that our

computed fold changes <1 correspond to decreased cell viability independent of cell viability issues resulting from DNA breaks. This is why we utilized pgRNAs that target non-expressed genes, rather than simply using non-targeting pgRNAs, as negative controls.

In addition to allowing us to control for the deleterious effects of DNA damage, utilizing a large set of negative control pgRNAs that targeted non-expressed genes allowed us to rigorously estimate an empirical false discovery rate for our screen. In brief, we calculated a p value for each target relative to all control pgRNAs. We then computed an empirical FDR by subsampling groups of 9 control pgRNAs from all pgRNAs targeting unexpressed genes 10,000 times to build an empirical distribution of p values generated from fake targets (e.g. subsampled groups of 9 control pgRNAs). Finally, we compared the computed p value per target to the p value distribution arising from the subsampling procedure in order to generate an empirical FDR per target.

5. PolyA-seq following pKO with the pgRNA.

Forced distal PA usage in ATG7 did reduce growth in vivo but did not increase p62 levels - were the authors able to show by PolyA-seq the expected change in length of 3'UTR following pKO with pgRNA, to help rule out off target effects of these pgRNA as an explanation for the observed phenotypes?

REPLY: We thank the reviewer for this comment. We followed the reviewer's suggestion and performed Poly(A)-seq for all polyclonal cell lines generated for this manuscript, including for *Atg7* pKO cells. This analysis revealed on-target, specific 3' UTR elongation of the targeted gene in each instance (e.g., see plots below for *Atg7* and *Egln1* pKO Poly(A)-seq compared to the control).

We also completed genome-wide differential APA analysis using these data to find that *Atg7* displayed the largest effect-size genome wide and showed significant 3' UTR lengthening, consistent with on-target effects of the *Atg7*-targeting pgRNA.

6. ATG7 follow-up studies

Autophagy was ruled out as a mechanism of action of the ATG7 3'UTR lengthening on tumour cell growth in vivo using p62 expression and a LC3 reporter – are these sufficient to confidently exclude an autophagy effect here as a reason for the observed phenotype? Otherwise, we're left with changes in cell cycle, which is not a particularly satisfying mechanism of action.

REPLY: We thank the reviewer for this comment. Like the reviewer, we were surprised to not find any detectable autophagy deficits in these cells despite performing multiple assessments of autophagy.

Since our previous assays focused on induction of autophagy through starvation (**Supp. Figure 10**), we performed additional autophagy assays to assess if autophagy inhibition might reveal an autophagy phenotype in *Atg7* pKO cells. We used the chemical inhibitor Bafilomycin A1 to assess cell sensitivity to autophagy inhibition. We exposed cells to increasing concentrations of Bafilomycin A1 and found that *Atg7* pKO cells were similarly sensitive to autophagy inhibition as were control cells. In contrast, *Atg7* KO (full gene knockout) cells were significantly more viable than were control or *Atg7* pKO cells at high Bafilomycin A1 concentrations. These new data are consistent with the data in our original manuscript and support our conclusion that B16-F10 cells treated with *Atg7* pKO pgRNAs still have functional autophagy in all assays we completed.

Although it remains possible that there are more subtle, context-specific alterations in autophagy that our experiments to date have not uncovered, all of our data thus far are concordant and suggest that *Atg7* pKO does not significantly impact canonical autophagy. We agree that identifying changes in autophagy would have been very satisfying, but the cell cycle alterations that we identified in *Atg7* pKO cells appear to be the most likely mechanistic explanation for the reduced cell growth that we identified in *Atg7* pKO cells.

Beyond the above experiment, the revised manuscript includes important new ATG7 follow-up studies demonstrating that restoration of ATG7 protein rescues the reduced growth phenotype that we observed in *Atg7* pKO cells (in which we ablated the gene-proximal poly(A) site):

- As the *Atg7* pKO cell line is a specific knock-out of the short *Atg7* isoform, we used this cell line as a background for our rescue experiments. We generated two *Atg7* cDNA constructs that harbored identical *Atg7* coding sequences and either an SV40 poly(A) signal directly downstream of the stop codon or the endogenous *Atg7* proximal 3' UTR including 200 nucleotides after the proximal poly(A) signal sequence (see schematic below and **Supp. Figure 8A**). We then validated that stable integration of the cDNA constructs rescued the reduced protein level present in *Atg7* pKO cells (**Supp. Figure 8B**). We then measured cell growth in vitro and found that introduction of either the *Atg7* SV40 or *Atg7* short 3' UTR cDNA rescued the reduced cellular proliferation phenotype present in *Atg7* pKO cells (**Supp. Figure 8C**).
- We observed complete rescue with the SV40 construct and partial rescue with the short 3' UTR construct. As the SV40 construct led to higher ATG7 protein levels, these data are consistent with a model where reduced ATG7 protein levels, rather than loss of the

short 3' UTR isoform per se, in *Atg7* pKO cells are the primary driver of the reduced cell growth phenotype of those cells (with the caveat, which is unavoidable in such rescue experiments, that transgenic expression of a gene in a rescue experiment is not fully equivalent to expression from the endogenous locus).

Transgenic restoration of ATG7 protein rescues the *Atg7* pKO growth phenotype.

(A) Schematic of two *Atg7* cDNA constructs harboring identical coding sequences, but distinct 3' UTRs, with either an SV40 poly(A) signal or the endogenous 3' UTR and the proximal poly(A) site.

(B) Immunoblot of protein collected from Cas9-expressing B16-F10 cells treated with either a control pgRNA or *Atg7* pKO pgRNA with stable expression of the indicated *Atg7* cDNA construct. Protein ratio normalized to α -Tubulin concentration and then to the control pKO protein ratio.

(C) *In vitro* cell growth of Cas9-expressing B16-F10 cells treated with a control pgRNA or *Atg7* pKO pgRNA with the indicated cDNA constructs as measured by CellTiter-Glo. Measurement is the average of three replicates +/- standard error of the mean.

Overall, these data strongly indicate that ablation of the *Atg7* proximal poly(A) site results in reduced cell growth due to reduced levels of ATG7 protein. While further work to mechanistically characterize the role of distinct isoforms of *Atg7* (as well as other targets

identified in our screen) in cancer would be interesting and valuable, we believe that the bulk of our work's innovation and novelty is the application of high-throughput functional screening to poly(A) site selection in the context of tumorigenesis. We hope that our study will help to motivate future studies that focus on the biological and functional roles of specific genes with APA events of high interest, including *Atg7*.

Reviewers' Comments:

Reviewer #1:

Remarks to the Author:

The authors have addressed my concerns.

Reviewer #2:

Remarks to the Author:

The authors have addressed my concerns.

Reviewer #3:

Remarks to the Author:

I thank the author and team for the thorough response to my comments and in particular:

the improved labelling of the supp table headers,

the description in the introduction of how polyA sites can lead to variable lengths of 3' UTRs,

clarification of why a median UTR length was as a parameter given the lack of normal tissue samples for most TCGA studies,

why the control pgRNA targeting unexpressed genes have such a high degree of effect variance (compared to control gRNA targeting coding regions),

and confirmation that autophagy is not the responsible for the effect of ATG7 UTR lengthening on tumour growth. The exact mechanism for this therefore remains unknown and I look forward to future publications that may explain this!

I hope that the inclusion of these additions has improved the clarity of the paper for the readers.